**Mechanisms of the Overturning Circulation in the Northern Red Sea, more than Convective Mixing**

Lina Eyouni[1,2], Zoi Kokkini [1,3], Nikolaos D. Zarokanellos [1,4], Burton H. Jones [1]

[1] King Abdullah University of Science and Technology (KAUST), Biological and Environmental Sciences and Engineering (BESE), Red Sea Research Center (RSRC), Thuwal, Saudi Arabia.

[2] Red Sea Global (RSG), Riyadh, Saudi Arabia.

[3] Consiglio Nazionale Delle Ricerche (CNR), Istituto di Scienze Marine (ISMAR), Lerici, Italy

[4] Balearic Islands Coastal Observing and Forecasting System (SOCIB), Palma de Mallorca, Spain

*Correspondence to*: Zoi Kokkini (zoi.kokkini@sp.ismar.cnr.it)

**Abstract.** The northern Red Sea (NRS) is where Red Sea Outflow Water (RSOW) and, occasionally, Red Sea Deep Water are formed. Glider observations are used to describe the formation mechanisms and pathways of the intermediate waters in the NRS in late winter from 31 January to 18 April 2019. Utilizing glider observations, atmospheric reanalysis products, and satellite datasets, we evaluated the mesoscale activity and the atmospheric conditions that contribute to outflow water formation. The cyclonic circulation in the region surfaces dense water, which exposes it to the atmosphere, ventilating the water column and contributing to phytoplankton growth (enhancement of chlorophyll concentration) due to the nutrients upwelled into the euphotic layer (Zeu). Subduction of this water in the 3-dimensional cyclonic circulation transported oxygenated, elevated chlorophyll water to depths between 150 m and 250 m along the 28.2 kg/m3 isopycnal. Unlike previous observations, in late February, a strong anticyclonic circulation blocked the inflow of warmer, fresher water into the region. It was accompanied by a negative heat flux and an uplifting of dense water to the surface. Net cooling through mid-March cooled the incoming surface waters from the south. At the end of the observational period, the intrusion of warmer, fresher waters from the south coincided with the re-establishment of cyclonic circulation and capped the dense surface water that had formed during March. These observations demonstrate that multiple processes

contribute to RSOW formation:  convective mixing, cyclonic uplifting of dense water, subduction, and

meso- (submeso-) scale processes.

## 1. Introduction

The Red Sea (RS) is a narrow, elongated, meridionally oriented basin lying between the Asian and African continents. Its subtropical location results in significant buoyancy losses due to high evaporation (nearly 2 m/yr), negligible precipitation, and effectively no riverine inputs, resulting in it being one of the world's saltiest and warmest seas (Edwards and Head, 1987; Smeed, 1997; 2004; Sofianos et al., 2002). The RS experiences seasonally reversing winds over the southern region, coupled with the Arabian Sea's monsoonal forcing. The reversing winds control the circulation and water mass exchange through the Strait of Bab al Mandab (Abualnaja et al., 2015; Patzert, 1974). These processes, along with the buoyancy forcing, drive the large-scale circulation (Bower and Farrar, 2015; Murray and Johns, 1997; Patzert, 1974; Sofianos and Johns, 2007). The inflow of water from the Gulf of Aden compensates for the evaporative water loss in the RS. The advected northward waters contribute to the overall latitudinal gradient in salinity and temperature from the south to the north. The northward advection of comparatively fresh and warm water from the south has an important role in the stratification (Asfahani et al., 2020; Churchill et al., 2014; Sofianos and Johns, 2007; Zarokanellos et al., 2017b), with the annual flux into the RS reaching up to 0.22 Sv (Sofianos and Johns, 2002). This water has been traced to the NRS at 28°N (Zarokanellos et al., 2017a). It is modified through heating, evaporation, and mixing as it progresses northward (Cember, 1988; Sofianos and Johns, 2003; Sofianos and Johns, 2007).

Significant mesoscale activity is found along the RS's main axis (Morcos, 1970; Morcos and Soliman, 1972; Quadfasel and Baudner, 1993; Sofianos and Johns, 2007; Zhan et al., 2014) that results from baroclinic instabilities (Zhan et al., 2014) and the presence of the Eastern Boundary Current (EBC; Zarokanellos et al., 2017a, 2017b; Biton et al., 2010; Biton et al., 2008; Bower and Farrar, 2015; Eshel et al., 1994; Eshel and Naik, 1997; Sofianos and Johns, 2007). Some of these mesoscale eddies are thought to be quasi-permanent features, in particular the cyclonic eddy (CE) in the Northern Red Sea (NRS) and the anticyclonic eddy (AE) in the Central Red Sea (CRS) (Chen et al., 2014; Yao et al., 2014b). The CE in the NRS plays a crucial role in forming the RSOW (Asfahani et al., 2020; Sofianos and Johns, 2007). Zarokanellos et al. (2017a) have shown that an AE in the CRS can redirect or deflect the advected northward flow of Gulf of Aden water. These mesoscale eddies substantially affect heat and salt advection and distribution of biogeochemical properties (Chen et al., 2014; Raitsos et al., 2013; Triantafyllou et al.,

2014), which are often useful tracers of physical processes. Eddies also transfer energy and momentum to the mean flow, driving the circulation (Lozier, 1997). In addition, mesoscale features fundamentally modulate local nutrient fluxes and phytoplankton dynamics (Longhurst, 2007). Eddy activity in the RS has shown that it significantly influences the optical and biological properties (Kürten et al., 2016; Mahadevan, 2015; McGillicuddy et al., 1998; Pearman et al., 2017; Siegel et al., 2008). In particular, mesoscale eddies in the RS facilitate the vertical transport of nutrients into the upper water column, thereby impacting biogeochemical variability (Zarokanellos et al., 2017; Kürten et al., 2019).

Mesoscale processes can induce subduction events over larger areas with significant impact in the marine ecosystem (Mahadavan et al. 2016). Subduction is the process where physical and biogeochemical tracers transport from the surface to the interior. This, in turn, influences the chlorophyll flux in the Zeu playing a significant role in the RS biological variability. Furthermore, while the upper ocean is primarily ventilated through subduction events, the ventilation in the deep ocean occurs primarily through open-ocean convection (Asfahani et al., 2020; Williams and Meijers 2019). Seasonal mixing, eddy interactions, and cross-shelf exchange drive biogeochemical fluxes in the upper layer. In the RS, seasonal phytoplankton variability has been studied primarily through remote sensing (Acker et al., 2008; Gittings et al., 2018; Racault et al., 2015; Raitsos et al., 2013), with only a few studies incorporating in situ observations (Kheireddine, 2020; Zarokanellos et al., 2017a, 2017b; Zarokanellos & Jones, 2021). These studies highlight ecological connections between coastal and open-sea regions, vital for coral communities (Acker et al., 2008; Raitsos et al., 2017). However, remote sensing is limited to surface observations, as ocean color data do not capture subsurface phytoplankton distributions.Integrating bio-optical data from both remote sensing and in situ measurements has provided valuable insights into the RS's biogeochemical dynamics (Brewin et al., 2015; Gittings et al., 2018; Kheireddine et al., 2020; Racault et al., 2015; Raitsos et al., 2013; Tiwari et al., 2018; Zarokanellos et al., 2017a, 2017b; Zarokanellos & Jones, 2021).

Water mass transformation typically occurs in the surface ocean in specific regions where air-sea interaction acts powerfully on the upper layer (Emery, 2001; Iselin, 1939). Significant surface cooling results in buoyancy loss that can contribute to deep mixing and convection. The NRS has been considered the main area of RSOW formation and, occasionally, Red Sea Deep Water (RSDW) formation

(Papadopoulos et al., 2015; Sofianos and Johns, 2003; Sofianos and Johns, 2007; Yao et al., 2014 b), due to the high evaporation rates and significant surface water cooling that occur during winter (Papadopoulos et al., 2013). These preconditions, in addition to the presence of the CE, where shallowing of the isopycnals occurs at the eddy center, contribute to the formation of the aforementioned RSOW and RSDW (Abualnaja et al., 2015; Asfahani et al., 2020; Sofianos and Johns, 2003; Yao et al., 2014 b; Zhai et al., 2015). The newly formed RSOW gradually sinks until it reaches an equilibrium density of near 27.5 to 27.7 kg/m$^3$ throughout the basin (Zhai et al., 2015) and flows southward, where it exits the RS through the strait of Bab-El-Mandeb (Cember, 1988; Sofianos and Johns, 2003; Yao et al., 2014 b). Once it enters the Gulf of Aden, it is mixed due to the intense mesoscale eddy activity at depths between 400 and 1000 m (Bower and Furey, 2010). Despite significant dilution of the thermohaline properties, RSOW has been observed as far southward as the Agulhas Current below 32˚S (Beal et al., 2000) and as far east as the Bay of Bengal (Jain et al., 2017).

Numerical simulations and a few in-situ observations suggest three potential mechanisms for RSOW formation. The first mechanism, open-ocean convection, is associated with the presence of a cyclonic gyre in the NRS (Sofianos and Johns, 2007; Papadopoulos et al., 2015). Strong atmospheric forcing in the region and the presence of cyclonic gyre creates favourable conditions for convection events. The second mechanism is mixed layer deepening during winter, resulting from a large negative heat flux (Zhai et al., 2015). The third mechanism is the combined effect of the cyclonic gyre and the weakening of the stratification that results from strong atmospheric forcing (Chen et al., 2014; Clifford et al., 1997; Manasrah et al., 2004; Morcos and Soliman, 1972; Yao et al., 2014 b). Concurrent with the formation of the RSOW, the RSDW forms in the NRS when strong cooling and evaporation occur in the gulfs of Suez and Aqaba (Cember, 1988; Papadopoulos et al., 2015; Sofianos and Johns, 2003; Yao et al., 2014 b). The RSDW is evident below 300 m, while the RSOW is typically found between 200 and 300 m and can be identified by the relatively high oxygen concentration, which can be distinguished from the RSDW. The regionally formed RSOW has a significant role in the overturning circulation, as indicated by in-situ observations and numerical simulations (Papadopoulos et al., 2015; Sofianos and Johns, 2003; Zhai et al., 2015). In addition, RSOW contributes to the ventilation of the RS (Papadopoulos et al., 2015; Sofianos

and Johns, 2007; Woelk and Quadfasel, 1996; Yao et al., 2014 b; Zhai et al., 2015) and the salt budget of
the Indian Ocean due to the high salinity concentrations present in intermediate depths (Beal et al., 2000).
The primary objective of this study is to understand the mechanisms that contribute to the mass
formation of RSOW in the NRS and the biogeochemical responses associated with these physical
processes. The second objective is to evaluate how atmospheric forcing affects mesoscale dynamics in
the study area. This paper is organized as follows: In Section 2, the in-situ, reanalysis, and remote sensing
observations are presented. Section 3 describes the coupling of the atmospheric forcing and the in-situ
observations. A comprehensive description of the flow variability and the mechanism of RSOW
formation is described, including the coupling between physical and biogeochemical processes in the
study area. Finally, Section 4 presents a discussion and comparison with previous regional studies, along
with the conclusions.

## 2. Data and methods

### 2.1 Glider observations

A sustained glider line that was traversed in a 3–4-day period in the NRS was used to capture the wintertime evolution of physical and biogeochemical characteristics in the NRS (Eyouni et al., 2024). The Seagliders (hereinafter "gliders") were equipped with a CTD, a dissolved oxygen sensor, and a triplet fluorometer (**Table 1**). The glider was deployed along a transect oriented approximately perpendicular to the coastline offshore from Duba, referred to as the "Duba line" (**Fig. 1a**). The line extended from approximately 5 km off the coast to nearly 75 km offshore. The glider completed each transect in about 3.5 days and was programmed to dive from the surface to ~750 m. The average horizontal speed was ~25 cm/s, and each dive cycle took ~3 hours to complete, depending on the target depth, topography, and sea conditions. The deployment spanned the period from 31 January to 21 April, 2019 (**Table 1**, **Fig. 1b**).

| Deployment Location | Observation Period | Duration (days) | Sensors | Parameter |
|---|---|---|---|---|
| NRS | 30.01.2019/22.04.2019 | 83 | Seabird, unpumped CT | CTD (Conductivity, Temperature, Pressure) |
| | | | WETlabs EcoPuck Fluorometer (FL3-IRB sensor) | Chlorophyll Fluorescence (CHL), Colored Dissolved Organic Matter (CDOM), Phycocyanin |
| | | | WETlab Backscatter (BB3 IRB sensor) | Backscatter at 532, 650, and 880 nm |
| | | | Aanderaa Optode (4330) | Dissolved Oxygen (DO) |

**Table 1:** Summary of the measured variables and their corresponding sensors deployed on the glider.

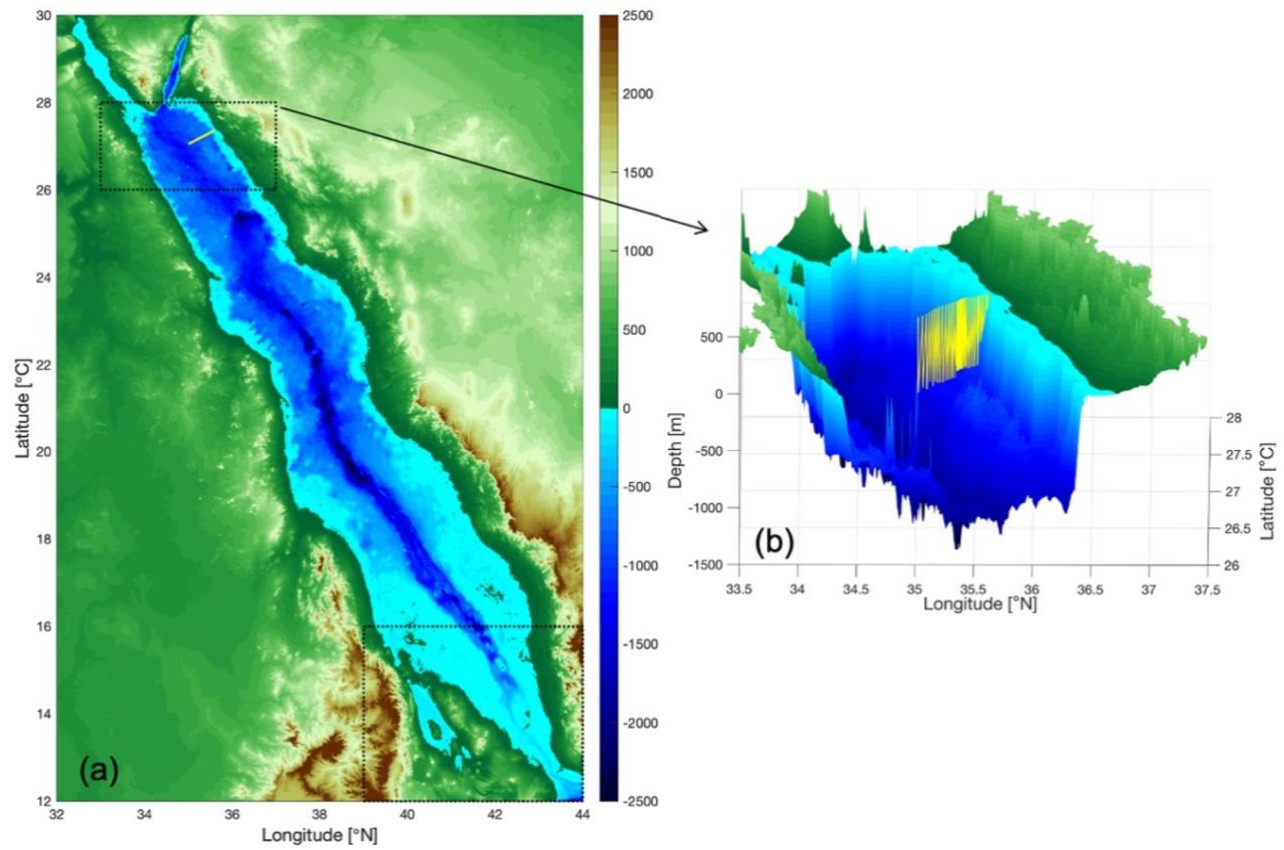

**Fig. 1:** Topographic map of the RS. The two black dash boxes in the northern and southern RS show the study area in the north and the region in the south where exchange with the Gulf of Aden occurs. The yellow line inside the northern box indicates the glider line location, and b) the yellow sawtooth line represents the undulating trajectory of the glider superimposed on the bathymetry, which has been reproduced from the GEBCO_2021 Grid, GEBCO Compilation Group (2021; https://doi.org/10.5285/c6612cbe-50b3-0cff-e053-6c86abc09f8f).

Each vertical profile has been examined for spikes and outliers before further analysis. No evidence of thermal lag has been observed in the CT observations. The WETlabs EcoPuck Triplet Fluorometer (FL3) and backscatter (BB3) sensor were factory calibrated, and dark counts were measured prior to

deployment to account for any drift. However, the BB3 sensor starts to drift on 26 March 2019. The pre-
deployment dark counts for the FL3 were consistent with factory dark counts. Roesler et al. (2017)
performed a global comparison of fluorometer and extracted CHL measurements and recommended that
factory-calibrated chlorophyll should be divided by two. Thus, CHL here is divided by 2 to correspond
to in situ CHL concentrations as described in Roesler et al. (2017). As the RS is a region with high
irradiance, the CHL can often experience quenching. In this study, we examined the vertical profiles of
CHL fluorescence for quenching, and no significant quenching was detected. The oxygen measurements
were adjusted by a correction factor based on the median oxygen saturation in the upper 10 m. The
correction factor was the ratio of the median oxygen saturation concentration for the upper 10 m to the
median measured concentration within the upper 10 m of the water column. The entire water column data
was then corrected by multiplying the reported oxygen concentrations by the correction factor. The
correction provided consistency between the current glider data set and the vertical distribution for this
region in the World Ocean Atlas (Zarokanellos and Jones, 2021; Garcia et al., 2019a). To facilitate data
processing, plotting, and analysis, the corrected data were projected onto a grid with a vertical resolution
of 2 m and a horizontal resolution of 2.5 km, the nominal distance between glider surfacing during the
mission. This study focuses on the upper 500 m of the water column. Mixed Layer Depth (MLD) was
calculated for each vertical profile based on a density difference of $\leq 0.03$ kg/m3 relative to the density
at 10 m depth (de Boyer Montégut et al., 2004). The Brunt-Vaisala frequency (BVF) was computed to
measure stratification. Geostrophic velocity and potential density were also calculated using the TEOS-
10 toolbox (TEOS, SCOR, and IAPSO, 2010). The level of no motion in the geostrophic velocity has
been examined, and no significant motion below 500 m has been observed.

**2.2 Remotely sensed data**
**2.2.1 Sea Level Anomaly (SLA)**
Previous satellite observations and numerical simulation studies suggest a cyclonic gyre located in
the NRS concurrent with convective mixing where the RSOW water mass is formed (Sofianos and Johns,
2002; Zhai et al., 2015). Therefore, in this study, SLA data were used to characterize the spatial and
temporal evolution of the large-scale circulation patterns of sea level and geostrophic velocity during the
wintertime. The SLA data is based on the multi-mission altimeter Archiving Validation and Interpretation
of Satellite Oceanographic Data (AVISO) provided by the Copernicus Marine Environment Monitoring
Service (CMEMS). The data has been gridded at a regular 0.258˚ X 0.258˚. The obtained daily measured
data were taken within the domain from 20˚ to 28˚ N and 32˚ to 42˚ E, inclusive of the glider deployments.
Within the NRS subdomain (26 - 28 ˚N, 33 - 37 ˚E), the data were temporally averaged to fit approximate
the periods of the glider transects. Previous work has validated the use of the AVISO product from
comparisons with both in situ and numerical model results (Hernandez and Schaeffer, 2001; Zhan et al.,
2014; Zarokanellos et al., 2017).

**2.2.2 Sea Surface Temperature (SST)**
Sea surface temperature (SST) data were obtained from the Moderate Resolution Imaging
Spectroradiometer (MODIS) imagery that provides a daily image with a spatial resolution of 4 km. The
data were obtained within the domain from 24˚ to 28˚ N to identify and determine the spatial and temporal
evolution of the large-scale patterns of the SST (Werdell et al., 2013).

**2.3 Atmospheric reanalysis products**

Atmospheric parameters for this study were derived from the Modern-Era Retrospective Analysis for
Research and Applications Version 2 (MERRA-2) reanalysis dataset (Rienecker et al., 2011; Gelaro et
al., 2017). MERRA-2 offers a high-resolution representation of atmospheric conditions, providing daily
mean values on a $0.5° \times 0.652°$ grid with a temporal resolution of one hour. This dataset was chosen due
to its demonstrated accuracy in representing heat fluxes within the region of interest, as validated by
recent studies such as Al Senafi et al. (2019). For this analysis, data were extracted and spatially averaged
over a defined box encompassing the NRS (NRS26 - 28 ˚N, 33 - 37 ˚E; **Fig. 1a**), excluding land coverage.
Daily means were calculated from the hourly data for the selected parameters of wind speed and direction,
surface net heat flux ($Q_{net}$), and evaporative heat flux, which is considered to be the primary driver of
heat loss in the RS (Sofianos ad Johns, 2003).
The $Q_{net}$, produced using the Coupled Ocean-Atmosphere Response Experiment (COARE 3.0)
formulation (Fairall et al., 1996), consists of the sum of shortwave ($Q_{sw}$, absorbed), longwave ($Q_{lw}$,
emitted), latent ($Q_L$, evaporative), and sensible ($Q_s$, conductive) heat fluxes, as all terms are positive when
they are heating the water column. The quantity $Q_{net}$ is calculated using the following formula:

$$Q_{net} = Q_{sw} + Q_{lw} + Q_S + Q_L \qquad\qquad (1)$$

**2.4 Empirical Orthogonal Function (EOF) analysis**
To compare the SLA from 2019 with observations from the preceding years, EOF analysis is carried
out on SLA data to evaluate the spatial and temporal patterns of the variability of the SLA data in the
winter-spring period. Each eigenvector describes the spatial pattern (modes) of that variability for five
months from January to May for the years 2016, 2017, 2018, and 2019. Only the first mode of the EOF
is used to compare the seasonal evolution of the spatial pattern each year. In addition, the explained
variance with the eigenvalue provides the relative contribution that a specific mode contributes to the
variability (Zhang and Moore, 2015).

**2.5 One-dimensional mixed layer model**
Price-Weller-Pinkel (PWP) mixed layer model (Price et al., 1986) has been applied to evaluate the
local atmospheric effects on the ocean mixed layer. The. The model input includes the following terms:
radiative heat flux (shortwave, longwave), latent and sensible heat, freshwater flux (evaporation [E] and
precipitation [P]; [E − P]), and wind stress components [$\tau_x$ and $\tau_y$]). As the RS is sandwiched between
two extreme desert regions, precipitation is considered to be negligible (P=0). The 1-D PWP has been
applied to estimate the local, atmospherically-driven evolution of the mixed layer depth during the
observation period. The model was executed for three sequential subsets delineated from the glider
observations: the cooling phase, the cool, salty, dense phase, and the warming-freshening phase. The
model was initialized with the average temperature and salinity profile for one complete glider transect
at the beginning of each simulation period (30 days) and then stepped forward in 24-hour (1-day)
increments subject to the heat, freshwater, and momentum fluxes. The daily time step was selected as
insignificant diurnal variability was observed in mixed layer temperature and salinity from the glider.

The PWP model produces a mixed layer through a vertical exchange process between the air and sea

interaction and vertical mixing. It assimilates time series of surface heat flux, wind, and precipitation and
applies these forcing parameters to the initial vertical profile of temperature and salinity. The model
interpolates the momentum components driven by winds, cooling, and evaporation to induce convective
instability, entrainment from the pycnocline, and a mixing term generated from vertical current shear. In
our case, since P=0, only surface heating affects re-stratification.

The convective adjustment in the PWP model starts with grid cells with unstable stratification being

homogeneously blended with neighbouring cells. The convective correction follows the bulk mixed layer
parameterization, where the mixed layer deepens when the bulk Richardson number, $R_{ib}$, falls below a
threshold value of 0.65 (Price et al., 1986).

The bulk Richardson number is expressed as: $R_{ib} = \frac{\Delta \rho g}{\rho_0 (\Delta U)^2}$              (2)

where $\Delta \rho$ (density) and $\Delta U$ (*velocity*) are the differences between their values within the mixed layer,
and their values below the mixed layer, respectively (Price et al., 1986). The variable $\rho_0$ is the reference
density and $g$ is the gravitational acceleration. Then, the model adds local shear instability below the
mixed layer, where mixing due to strong shear is parameterized based on a threshold gradient Richardson
number, $R_{ig}$, defined as: $R_{ig} = \frac{N^2}{S^2} = 0.25$                    (2a)

where, $S^2$ is the square of the shear below the mixed layer and $N^2 = \frac{g}{\rho}\frac{\Delta \rho}{\Delta z}$              (2b)

The vertical resolution of the model depth bin was set to 2 meters to be aligned with the gridded glider

data. The momentum (horizontal diffusivity) and vertical diffusivity were set to $10^{-5}$ m$^2$/s and 0 m$^2$/s,
respectively (Sanikommu et al, 2020; Zhai et al., 2015). The maximum depth of the PWP experiment for
the run and the initial depth range for the profiles of salinity and temperature were 400 m, the depth at
which the divergence of hydrographic variables between summer and winter was minimal. Estimation of
the MLD at each time step throughout the PWP model run used the same criteria used for the glider data
by determining the depth range over which the density increase relative to 10 m was no more than 0.03
kg/m$^3$ (de Boyer et al., 2004).

## 3. Results

### 3.1 Atmospheric forcing

Regional atmospheric forcing is a major factor affecting the seasonal variability of the RS circulation. The wind direction in the northern part of the RS is predominantly from the north-northwest (NNW; **Fig. 2a**). The net heat flux was initially negative, with heat flux losses of up to 300 w/m$^2$ in January, up to 250 w/m2 in February, and transitioning in late March from net negative to net positive, thus beginning the onset of the seasonal heating period (**Fig. 2b**).

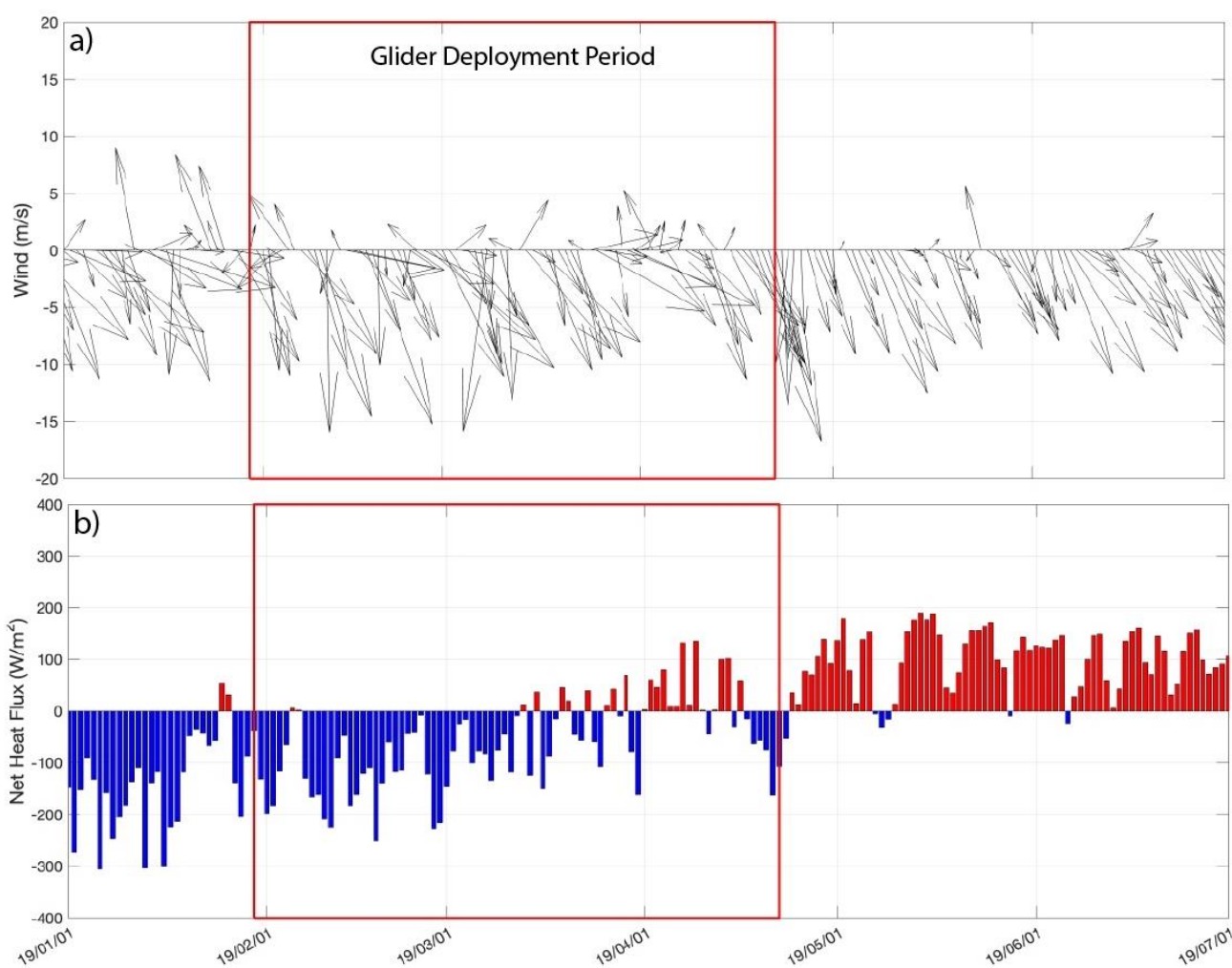


**Fig. 2:** a) Wind vectors for the NRS and b) net heat flux for the NRS (average for the black dashed box
over the NRS, **Fig. 1**) for the period from January 1, 2019 through July 1, 2019. The red box indicates the
period of the glider mission.

**3.1 Upper Ocean response to atmospheric forcing**
**3.1.1 Regional response from a remote sensing perspective**
Remotely sensed ocean imagery demonstrates the seasonal evolution of the upper ocean during the
glider deployment. **Fig. 3** provides images of 8-day composites for SLA and SLA-derived geostrophic
velocity, sea surface temperature, and CHL concentration. At the onset of the glider deployment in late
January–early February, the eastern boundary coastal flow was northward (**Fig. 3a**). Consistent with this
northward flow, a tongue of warmer, low-CHL water extended northward along the Saudi coastline (**Figs.**
**3b–c**). These observations are consistent with previous observations from 2016 (Asfahani et al., 2020)
and with the mean structure typically observed for SST in the winter months (Karnauskas and Jones,
2018). Cooler, higher CHL water is observed on the western side of the basin, perhaps due to the
convective mixing described by Kheireddine et al. (2020).
Following the initial phase of northward coastal flow, an AE developed in the northeastern RS (**Fig.**
**3d**). During this period, the flow was southward across the glider line, and there was no indication of
northward advection of warmer, low-CHL water (temperature > 24°C and CHL < 0.1 mg/m³) from the
south (**Figs. 3e-f**). The AE appeared to block the warmer, low-CHL water transport into the region. The
surface temperature of the NRS became cooler reaching a mean surface temperature near 22.5 °C and the
temperature difference between the western and eastern sides of the northern region decreased to less than
0.25 °C (**Figs. 3h** and **4a**).
In the latter half of March, two anticyclonic eddies (**Fig. 3g**), between 22 °N and 26 °N, apparently
blocked the northward flow of water from the south and isolated the northern part of the RS from the
inflow of the warmer, low-CHL water. Consequently, the near-surface in the NRS became almost
thermally homogeneous, with small temperature variations in its spatial distribution (**Fig. 3h**). The densest
surface water was observed in the NRS during this period (**Fig. 4c**). By the beginning of April, these two
eddies had dissipated, and warm, low CHL water again advected into the NRS along the eastern coastline
(**Figs. 3j**, **3k**, and **3l**), re-establishing the temperature differential between the eastern and western halves
of the NRS (**Figs. 4a-c**).

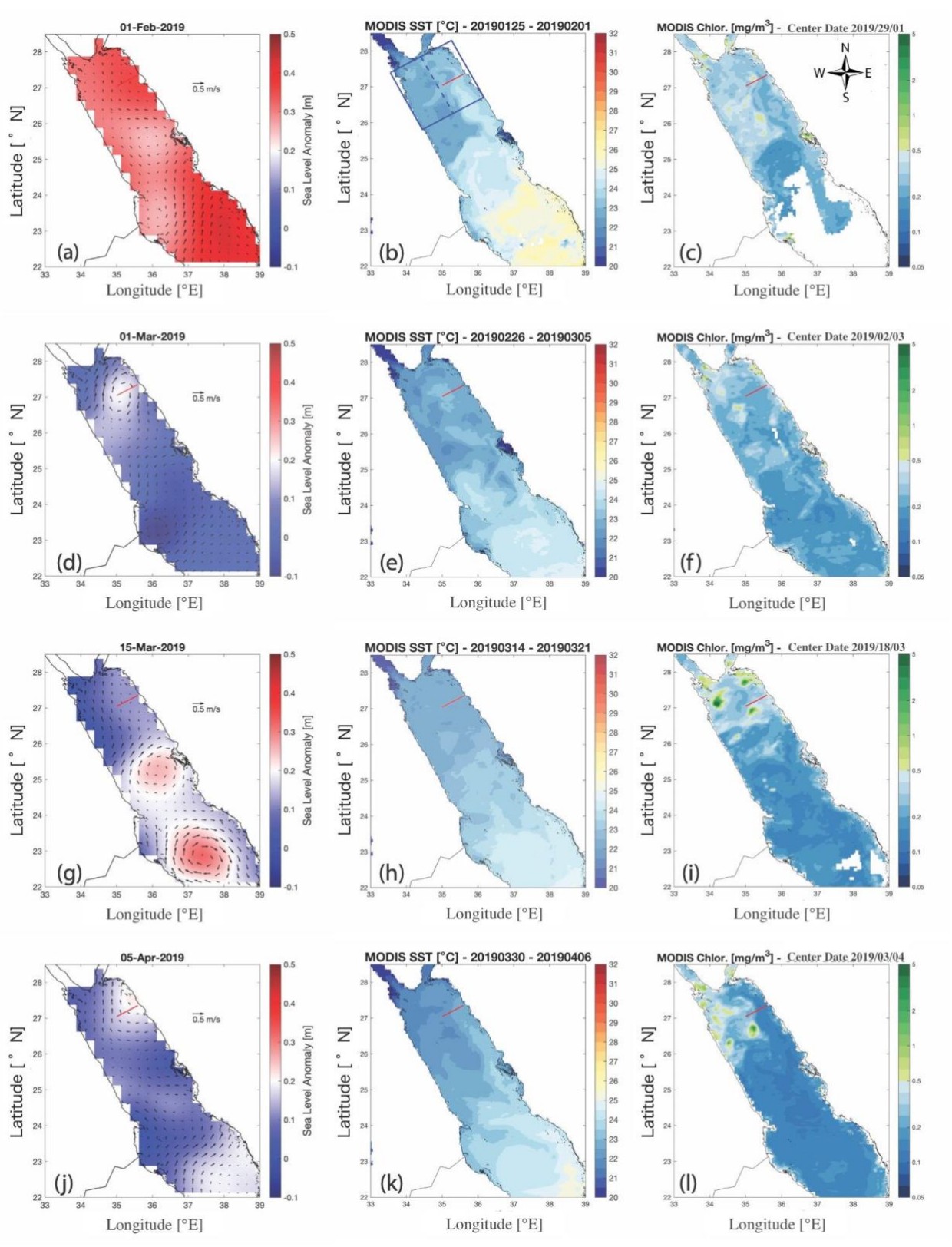

**Fig. 3:** 8-day averages of sea level anomaly and geostrophic velocity (left-hand panels) from AVISO as
provided by the Copernicus Marine Environment Monitoring Service (CMEMS), 4 micron nighttime SST
from the MODIS Aqua satellite (center panels), and CHL from MODIS OCI (right-hand panels) during
the presence of the EBC (panels a-c), AE (panels d-f), pair of anticyclonic eddies (panels g-i), and lateral
advection (panels j-l). The red line indicates the location of the Duba glider track. The blue box in panel
b) indicates the region of the northern RS (NRS) that was used for regional averages (~ 200 km x 200
km). The dashed blue line divides the eastern half of the NRS from the western half.

### 3.1.2 Upper layer variability

As the atmosphere progressed through its typical annual cycle, the near-surface ocean demonstrated high variability. In order to compare the data accurately, the depth of 6 m has been chosen to represent the near-surface layer, while the depth of 500 m has been selected to represent the near-bottom layer, as it is the most isolated from the surface influence and shows the least variability within the data set. The time series of surface (6 m) and 500 m values for temperature, salinity, and density are shown in **Fig. 4**. This is a continuous time series of glider data, irrespective of its location along the transect. Distinct phases are evident in the time series. The early phase, consistent with the atmospheric forcing, shows a general cooling trend from mixed layer temperatures near 24 ˚C in the early period to about 22.5 ˚C during the coolest phase, after which temperatures rose again to nearly 24 ˚C. For most of the period, the western half of the northern RS was cooler than the eastern half, where the glider was operating (**Fig. 4a**). However, during the coolest period, the temperatures were nearly homogeneous across the entire northern RS. Correspondingly, mixed layer salinities rose from values of 40–40.2 during the early phase to nearly 40.4 during the cool, salty period, then returned to values between 40 and 40.2 in the warming period. As a result, the near-surface density anomaly initially increased from 27.6-27.8 kg/m$^3$ to 28.3-28.4 kg/m$^3$ during the dense period. As expected, no effects of the surface forcing were apparent at 500 m.

In the latter part of the cooling phase (between February 17 and March 12), the MLD exceeded 100 m much of the time. Although the deepest mixed layer occurred during the cool, salty phase, the MLD during this period was highly variable (**Fig. 4d**). The variability in the MLD is a result of the new, well-formed cyclonic eddy during that time. Shallow MLD can be present in the center of a newly-formed cyclonic eddy, as is shown in **Fig. 7**, and it is also consistent with the patchiness of the SST (**Fig. 3e**).

Individual glider sections provide insight regarding mesoscale processes occurring in the NRS during this winter-spring period. In the early cooling phase, the isopycnals were tilted strongly downward from offshore to nearshore, with cool, dense water near the surface offshore and warmer, fresher water near the surface nearshore (**Figs. 5a** and **5b**). The overall geostrophic flow was northward, with an average velocity of 0.31 m/s in the upper 100 m (**Fig. 5e**), consistent with the geostrophic velocity calculated from the sea level anomaly (**Fig. 3a**). The maximum upper layer stratification occurred in the offshore 20 km of the transect, where uplifting of the pycnocline resulted from the cyclonic circulation (**Fig. 5d**). The

maximum near-surface CHL concentration occurred where dense water (≥28.1 kg/m3) offshore
shallowed to within 50 m of the surface.

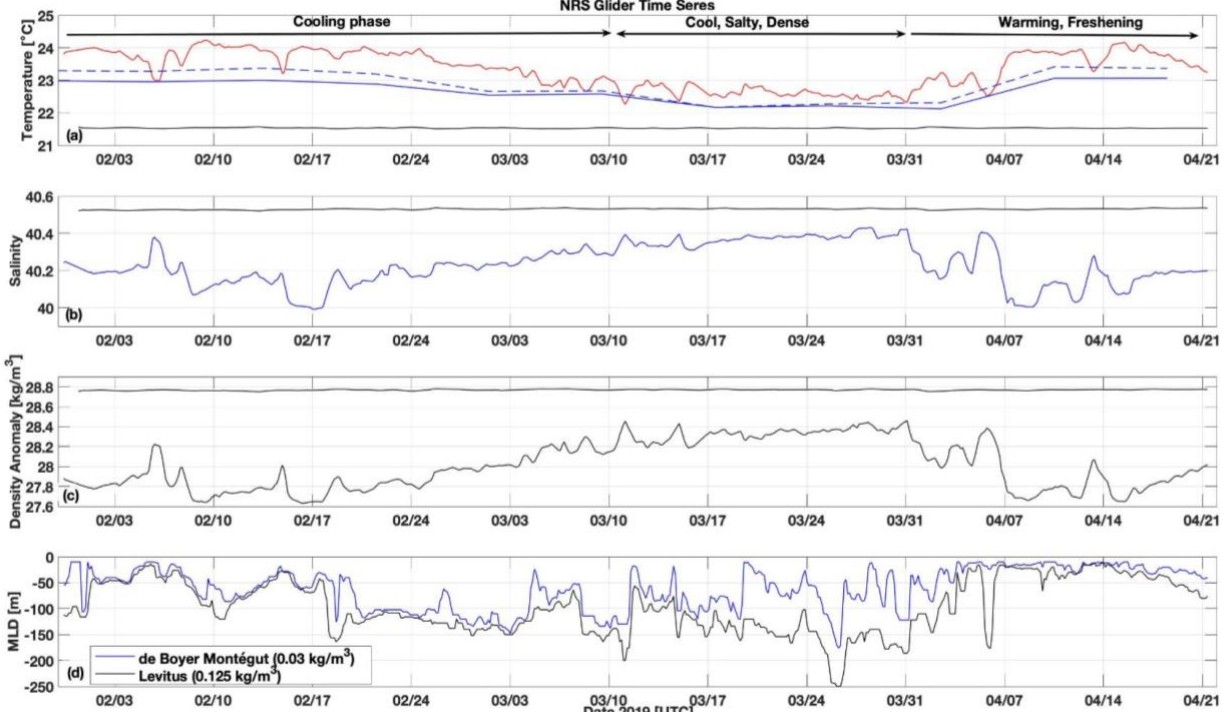


**Fig. 4:** Time series of near-surface (6 m) and deep (500 m) temperature, salinity, density, and MLD for
the glider deployment from January 31 through April 21, 2019. This is the complete time series,
irrespective of the glider's location along the transect. In panel a, temperature averages from the 8-day
MODIS SST are shown for the entire NRS (solid blue line), the eastern half of the northern RS (NRS-
East, dash blue line), and the western NRS (NRS-West, dotted blue line). The geographical boundaries
of these subregions are shown in **Fig. 3b**.

A small-scale cyclonic feature centered about 43 km offshore was embedded within the larger-scale

flow (**Figs. 5a**, **5b**, and **5e**). This feature was not observed in either the previous or subsequent transect,
each separated by approximately 3 days from the current transect. Thus, it was a transient feature on the
glider line, which we conclude was advected across the glider line within the larger scale flow. A feature
characterized by elevated CHL (∼30 × 10⁻³ mg/m³) and DO (∼178 μmol/kg) was observed along the
28.2 isopycnal, between 20 and 40 km offshore, compared to the rest of the transect where CHL was $\sim$
$5 \times 10^{-3}$ mg/m³ and DO $\sim$94 μmol/kg, respectively, while low BVF is observed in the same area. The
feature spanned at a depth range from approximately 150 m at 40 km offshore to 250 m at 20 km offshore,
suggestive of subduction of denser near-surface water and downward transport along the isopycnal below
the mixed layer and the euphotic zone (**Figs. 5c, 5d, and 5f**). This signal was also present in the
backscatter at 650 nm (Fig. 5h). While the higher concentration on the backscatter is evident offshore at
the surface ($\sim$4.5 x $10^{-4}$ $m^{-1}sr^{-1}$) and decreases with depth and proximity to the shore, in the same area
as the elevated CHL and DO, the backscatter reaches values $\sim$3 x $10^{-4}$ $m^{-1}sr^{-1}$ , whereas in the
surrounding waters it is less than 2 x $10^{-4}$ $m^{-1}sr^{-1}$. This aligns with subduction of CHL (**Fig. 5g**) and
backscatter at 650 nm (**Fig. 5h**) along the isopycnals between 28.2-28.3 $kg/m^3$ below the mixed layer and
Zeu (120 m) at depths greater than 200 m, contributing to the export of organic matter as it has been
observed in other regions of the global ocean (Zarokanellos et al., 2022; Zarokanellos and Jones, 2021;
Erickson et al., 2016)

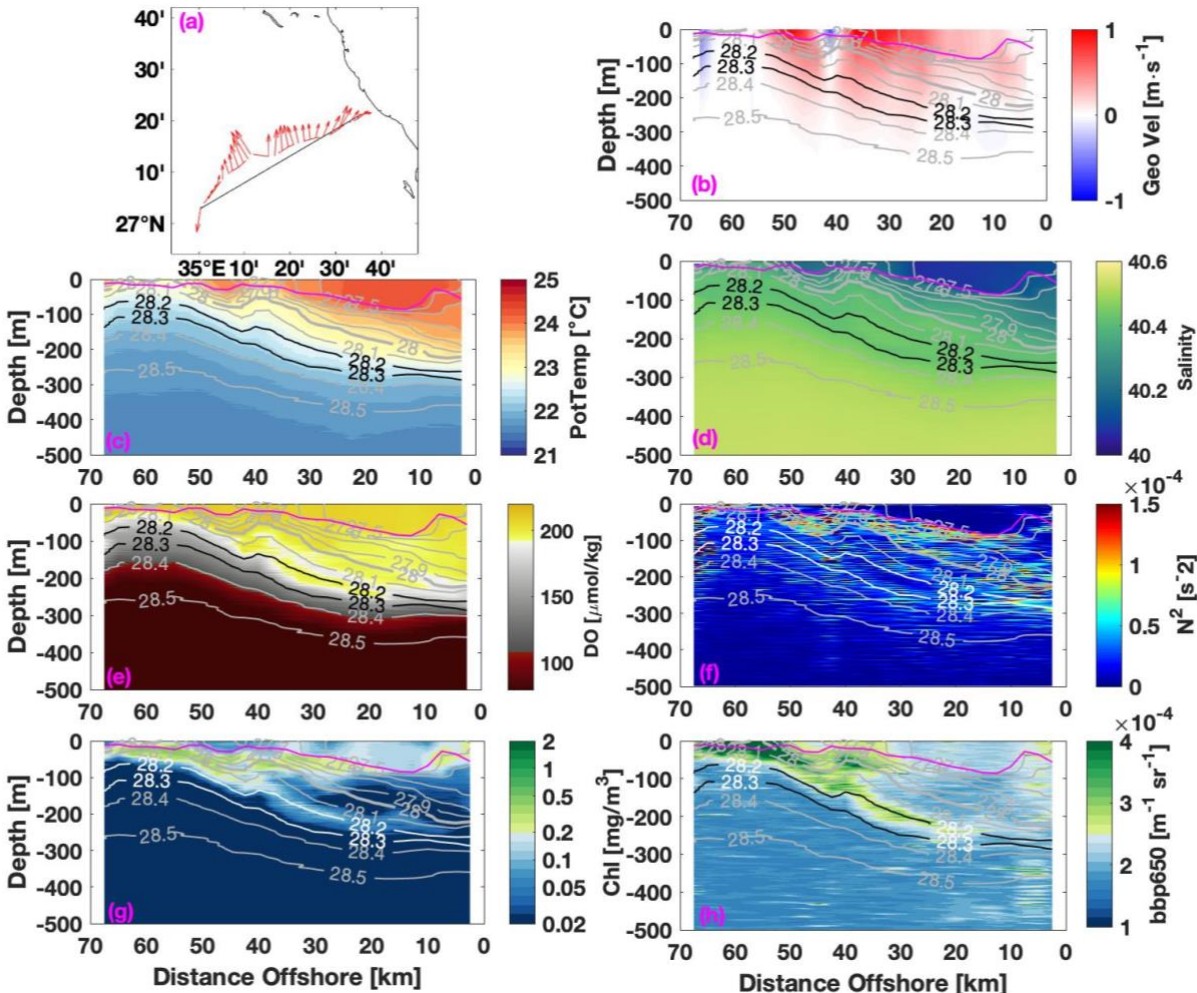


**Fig. 5:** Glider section of a) Depth Averaged Current (DAC), b) Geostrophic velocity, c) Potential temperature, d) Salinity, e) Dissolved Oxygen, f) Brunt-Väisälä frequency, g) Chlorophyll-a, and h) Backscattering coefficient at 650 nm, for February 5–9, 2019. Density isopycnals are shown in subfigures b, c, d, e, f, g, and h with solid grey lines, while the isopycnals of 28.2 and 28.3 are distinguished with black lines (or while for f, g, h). The solid magenta line represents the MLD. Geostrophic velocity, calculated relative to 500 m, is positive to the north-northwest and negative to the south-southeast parallel with the coastline.

The glider observations of CHL, backscatter at 650, and DO allow us to independently trace the subduction with three different bio-optical tracers. Indeed, the observed elevated CHL—commonly associated with phytoplankton growth in the literature—primarily occurs only offshore, within the Ze, where dense water ($\geq$28.1 kg/m3) rose to a depth shallower than 50 m, bringing up nutrients from deeper layers. Also, the Zeu is located around 120 m in the RS, and light at greater depths is too low to sustain photosynthesis (Zarokanellos and Jones, 2021). Furthermore, this transient eddy about 43 km offshore was not observed in either the previous or the following section, and it was embedded within the larger-scale flow (**Figs. 5a, 5b, and 5e**). The observed high DO concentration on the surface can be a result of photosynthesis. The co-occurrence of high CHL and DO at depths below the Zeu suggests that this water was originally at the surface before it transferred and subducted deeper. The fact that the high CHL and DO waters align along the 28.2 isopycnal (**Figs. 5, 11a, and 11b**) indicates that their subduction is associated with an eddy wherein the denser surface water is forced below, with the lighter water following the 28.2 isopycnal rather than being vertically mixed

Following the initial period of northward transport, the circulation changed significantly in late February, reversing the direction of flow across the glider line due to the presence of an AE in the north-eastern RS (**Figs. 3d-f** and **6**). Based on glider sections, the southward coastal flow began in mid-February and persisted for approximately 3 weeks. The isopycnal structure associated with the anticyclonic geostrophic flow is evident below the mixed layer. In the glider section from March 1–5 the geostrophic velocity varied between weakly northward at the offshore and inshore ends of the line to a maximum southward velocity of 0.67 m/s southward, and an average southward velocity of 0.14 m/s in the upper 100m, weaker in magnitude than the northward flow during the preceding cyclonic period (**Fig. 6b**). During this period, the upper boundary of the pycnocline, 28 kg/m3, which had been near the surface offshore during early February, was now at nearly 150 meters depth in the offshore portion of the transect but rose to the surface near the coast (**Fig. 6**). The MLD was consistently deeper than 100 meters in the offshore 30 km of the transect, and in the near-shore 40 km, it varied between ~25 and 120 m. CHL was uniformly distributed within the mixed layer with diel patterns in concentration, which appear as spatial patterns in the 3.5-day transit of the line (**Fig. 6g**).

Typically, the oxygenated waters are located in the surface layers within the MLD. However, the
observed bolus indicated that high oxygenated waters had trapped below the MLD, between the 28.2 and
28.3 kg/m³ isopycnals at 150 to 250 m depths and between 20 and 50 km offshore. The average DO
concentration within the bolus is ~177 µmol/kg, while CHL is around $4.6 \times 10^{-3}$ mg/m³. The surrounding
waters below the 28.3 isopycnal indicate that the DO and CHL values reach 62 µmol/kg and $2.9 \times 10^{-3}$
mg/m³, respectively. Above the 28.2 isopycnals, the DO and CHL have values of 203 µmol/kg and $79 \times$
$10^{-3}$ mg/m³, correspondingly. Compared to the underlying layers, CHL within the bolus is slightly
elevated (~3.6%), while DO is significantly higher by approximately 285%. The thickness of the layer
between these two isopycnals varies, ranging from less than 40 m, and the thickness of the trapped bolus
is approximately 100 m, indicating a distinct water mass that is also associated with low BVF. The
observed elevated BVF around the bolus suggests that this is a stable water mass isolated from the
surrounding water column rather than a result of vertical mixing. This lens is slightly warmer (~22.3°C)
and more saline (~40.43) than other waters within the same isopycnal range along the transect (**Fig. 6c,**
**6d, 6f**). While its signature was not reflected in CHL (**Fig. 6g**), the bolus is also detectable in backscatter
(**Fig. 6h**), with a concentration nearly 11% higher than the surrounding waters (**Fig. 6h**). This bolus is
likely outflow water from the Gulf of Aqaba, which might be advected into the region by the southward
flow and subsequently captured and recirculated by the observed AE (**Fig. 6a**). Only a few studies are
available regarding the water mass characteristics of the Gulf of Aqaba (Manasrah, 2002; Manasrah et
al., 2004), suggesting that the upper 300 m of the Gulf exhibit conditions similar to those found in the
upper 100 m of the NRS during winter, with temperatures ranging from 20.4°C to 22.4°C and for the
salinity between 40.3 and 40.7.
The near-surface temperature continued to decrease through March, while salinity increased within
the surface layer, reaching a maximum salinity of 40.4 and a density anomaly of about 28.4 kg/m³ in late
March (**Fig. 4**). The corresponding glider section of temperature and salinity for this period (**Figs**. **7c-d**),
March 26–29, shows that the coolest, saltiest, and densest water occurred in the center of a cyclonic eddy
about 40–45 km offshore, where the dense isopycnals (>28.2 kg/m³) outcrop at the surface. Across this
transect, the near-surface temperature was less than 22.8 ˚C, the minimum salinity was more than 40.3,
the minimum near-surface density was >28 kg/m³, and the minimum stratification was $<1 \times 10^{-4}$ (**Fig. 7f**).
Prior to this, the shallowest that the 28.2 kg/m³ isopycnal was observed was in the early section between
February 5 and 9, at a depth of about 90 m at the offshore end of the transect. In the same transect, the
isopycnal descended below 250 m nearshore. Mixed layer depth along this transect ranged from as
shallow as 12 m to as deep as 148 m on the inshore end of the transect, on the periphery of the eddy
circulation. In the eddy center, the mixed layer extended to the depth of the 28.3 kg/m³ isopycnal where
stratification due to the uplifted pycnocline impeded deeper mixing. The isopycnal uplifting appears to
be in the center of a cyclonic feature where the geostrophic velocity (**Fig. 7a, 7b**) is northward nearshore
(0.5 m/s) and southward offshore (0.2 m/s)

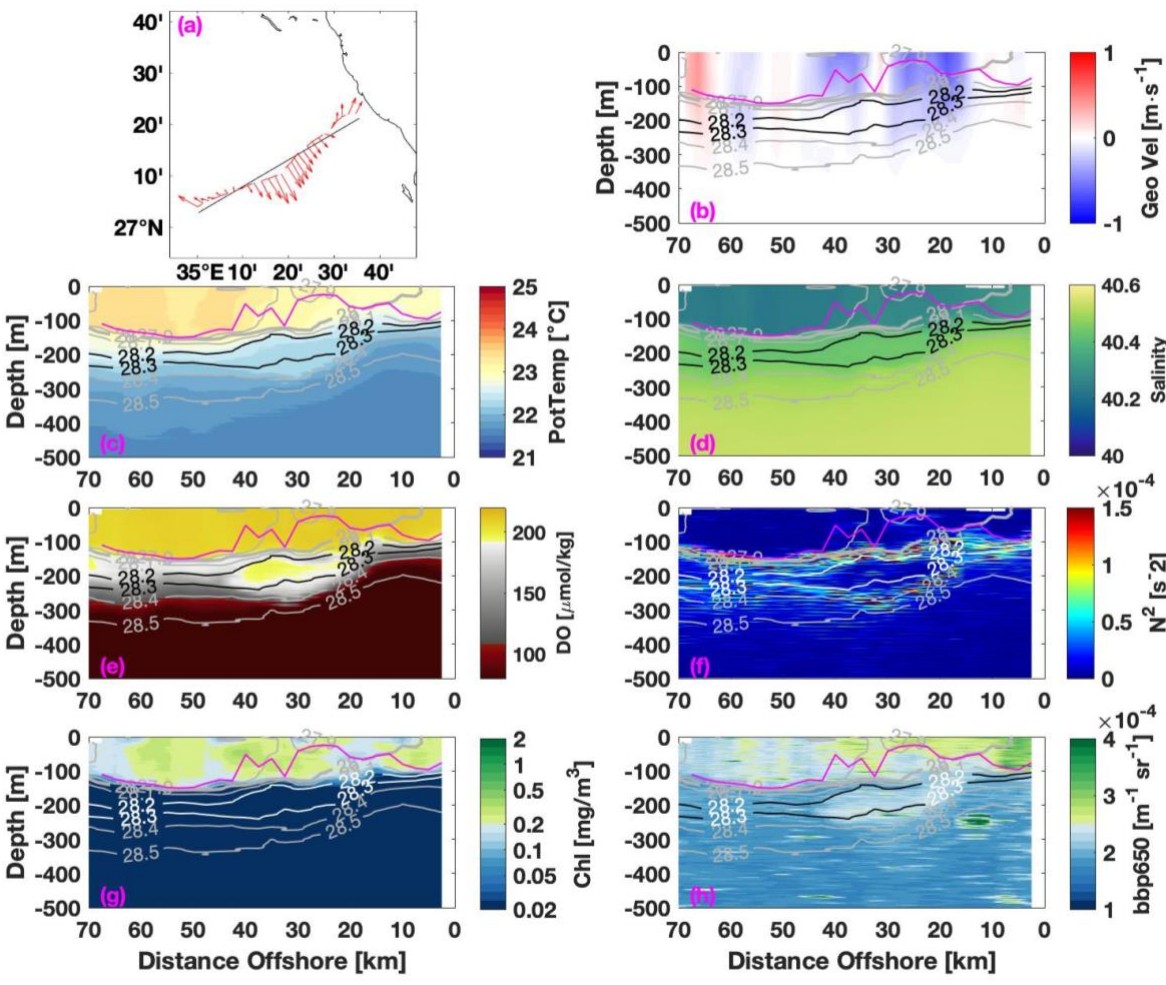


**Fig. 6:** Same as for **Fig. 5**, but for the period of March 1–5, 2019
As in earlier sections, small-scale structures are apparent and potentially important to biogeochemical
processes. At about 20 km offshore, a low DO and low CHL feature extends upward from about 200 m
to at least 100 m depth (**Figs. 7e-g**). The low DO concentration extends to shallower depths. The uplift of
isopycnals affects the biogeochemical processes by bringing low DO and CHL waters into the Zeu. This
process modulates nutrient, carbon, and DO availability and ultimately affects primary production.
Phytoplankton growth depends on the nutrients and light availability. The low-CHL waters typically
indicate nutrient-depleted conditions at the surface, while the low-DO waters in deeper layers are
generally enriched with remineralized nutrients such as nitrate, phosphate, and silicate (Garcia H.E. et al.,
2018). In this case, the low-CHL and DO waters have reached ～60 m, penetrating the Zeu, which extends
to ～120 m, as reported by Zarokanellos and Jones (2021). When these nutrient-available waters reach
the Zeu, they can stimulate phytoplankton blooms, enhancing primary production (Falkowski et al.,
1998). The uplift of the 28.3 isopycnal (～60 m) due to the presence of the cyclonic eddy (**Fig. 7**), also
influences nutrient availability (Zarokanellos & Jones, 2021; Kurten et al., 2019). This mesoscale eddy
activity in the region often drives the shift in the phytoplankton community (Kurten et al., 2019).
The presence of the CE leads to an uplift of the isopycnals about 50 km offshore (**Fig. 7**). In the
shoreward periphery of this eddy (~20 km offshore), both CHL and DO penetrate to a depth of up to 250
m, well below the mixed layer and the Zeu depth (**Figs. 7g** and **7e**). This subduction occurs near the
nearshore reversal of flow. Unfortunately, no backscatter data were acquired during this period (**Fig. 7h**).

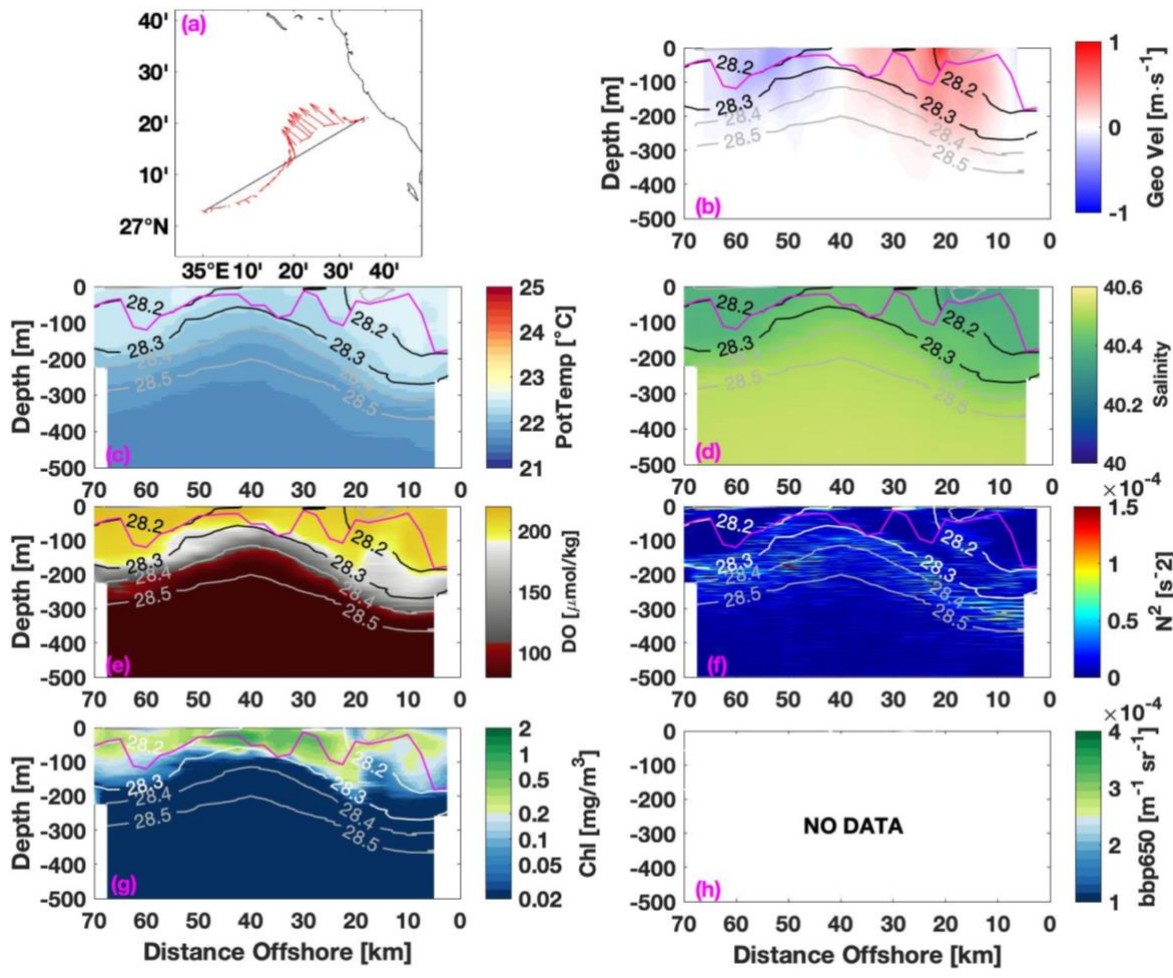

Fig. 7: Same as for Fig. 5, but for the period of March 26–29, 2019.

Immediately following this cool period, warmer, fresher water began to appear in the nearshore region of the glider line (Figs. 8c-d). The shallowing of lower-oxygen water, hence nutrient-rich, offshore between 35 and 50 km was observed in late March and subsided with the weakening of the cyclonic circulation (Fig. 8e). The cyclonic circulation remained (Fig. 8a), but it was entraining the warmer, fresher water from the south, as is evident in the composite SST image from March 30-April 6 (Fig. 3k). This warming and freshening continued through the remainder of the glider deployment (Fig. 4a). Freshening was evident on the nearshore half of the glider line, where salinities fell below 40.3, while

remaining higher along the offshore half of the line. Denser water subsided except at the outer half of the
transect. Although the direction of the geostrophic velocity was similar in pattern to the previous period
(**Fig. 7e**), the magnitude of the nearshore flow intensified by 0.2 m/s, while the offshore flow was similar
to the previous transect (**Fig. 8b**). **Fig. 8g** shows that the CHL concentration between 35 and 50 km
offshore decreased as warmer water was entrained near the coast.

Subduction is also present between 30 and 10 km onshore and is clearly observed in the BVF panel

and in CHL and DO (**Figs. 8b, 8e,** and **8g**). The subduction displaces the pigments of CHL and DO deeper
than 200 m nearshore; in the offshore area, these two parameters are well distributed in the first 100 m.

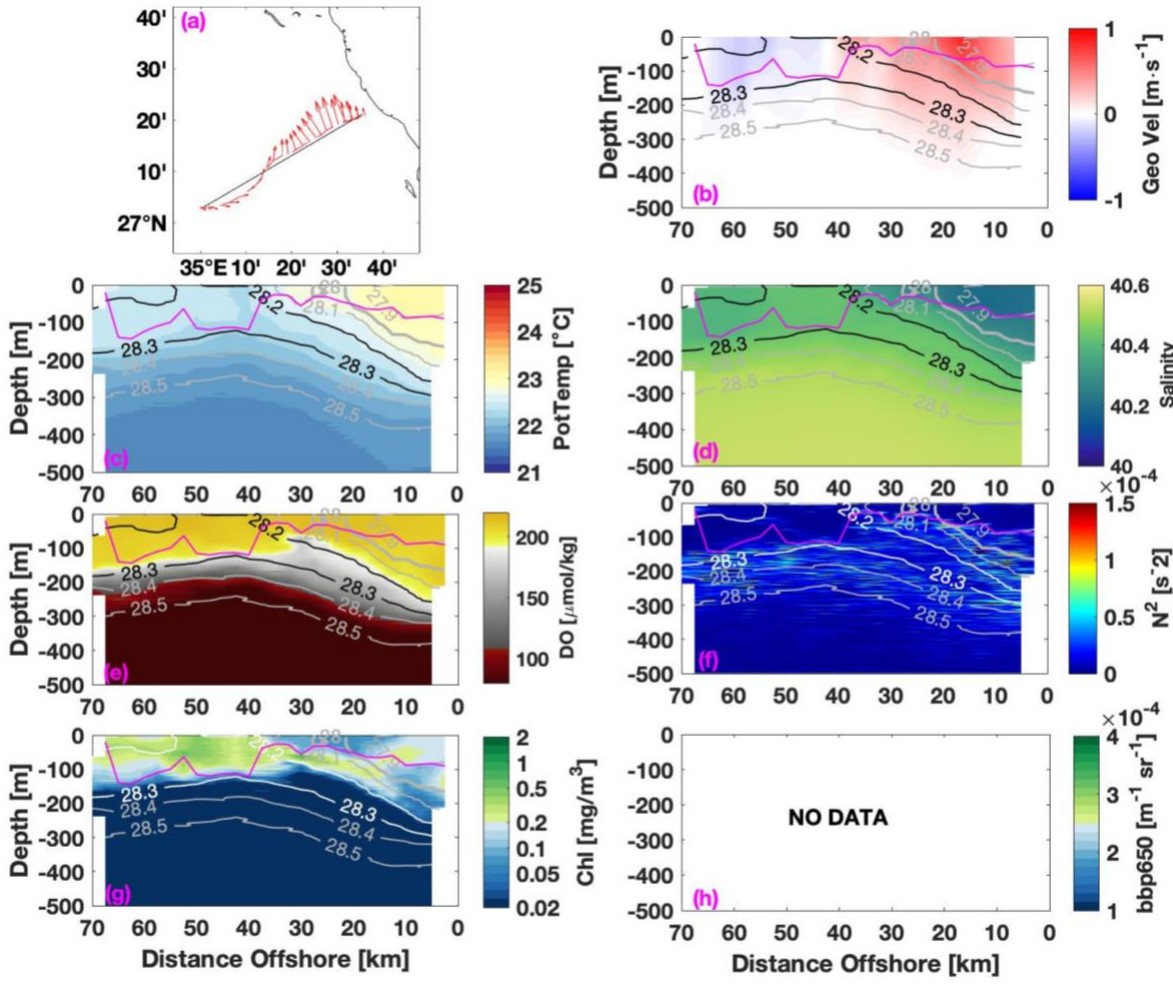


**Fig. 8:** Same as for **Fig. 5**, but for the period of March 29-April 2, 2019.

### 4. Discussion

The NRS is a dynamic and complex three-dimensional circulation with significant seasonal variability influenced by strong atmospheric forcing through wind stress and air-sea buoyancy fluxes. Direct observations and modeling experiments have both captured the formation of locally produced intermediate (RSOW) and deep water (RSDW) masses and their interactions with adjacent gulfs of Suez and Aqaba (Table 2; Asfahani et al., 2020; Sofianos and Johns, 2003; Papadopoulos et al., 2015). Two main thermohaline cells are associated with water mass formation and influenced by mesoscale dynamics, wintertime cooling, and deep convection. Numerical simulation studies suggest that the cyclonic gyre is the most probable site for RSOW formation (Yao et al., 2014a; Sofianos and Johns, 2003).

Unlike previous observations and interpretations of the NRS (e.g., Asfahani et al., 2020; Papadopoulos et al., 2015; Yao and Hoteit, 2018), this study observed a reversal of the currents in the eastern half of the basin prevented the inflow of warmer, fresher water from the south. During this phase, the upper layer of the NRS became relatively homogeneous, and near-surface water along the glider line reached its maximum salinities and densities (**Figs. 5d** and **7d**).

To evaluate the similarities and differences with previous years, an EOF analysis was performed on the SLA data between 26 ˚N and 28 ˚N over a period of 4 years that has been considered efficient (from 2016 to 2019; **Fig. 9**). The first mode of the EOF describes 86.6 % of the SLA variation (**Fig. 9a**). In the years 2016–2018, the EOF of the SLA showed a relatively positive or neutral pattern during the winter-spring transition period, which continued until May, when the EOFs typically decreased (**Fig. 9b**). This late spring decrease generally coincides with the transition from a net negative air-sea heat flux to a net positive flux (see **Fig. 2b**). For 2019, the first mode of the EOF showed a distinct increase in late January through mid-February, then became negative through mid-March, in contrast to the pattern in previous years. This negative phase is consistent with the period when the circulation was anticyclonic (**Fig. 9b**). The flow of warmer, fresher water from the south was apparently blocked during this period, and the temperature became relatively homogeneous in the NRS, as have been previously observed in the CRS (Zarokanellos et. al., 2017). Recently, a study by Mohamed and Skliris (2025) showed that the annual climatology of the sea level is generally higher on the eastern boundary of the RS compared with the western boundary, where many areas are isolated patches of higher or lower values of SLA indicate

mesoscale activity. The maximum values correspond with regions where AEs are present. In our study,
the negative phase of the EOF analysis (**Fig. 9b**) aligns with the presence of AE in the NRS.

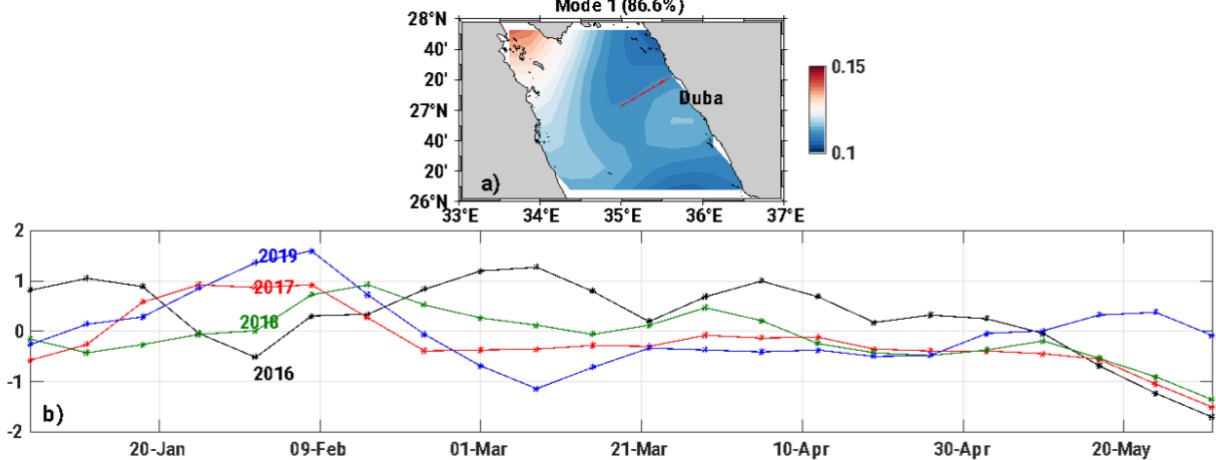


**Fig. 9:** The first mode of the EOF based on the SLA weekly mean in 2019. (a) the spatial pattern of the
first mode. (b) time series graph of the first mode for four (4) subsequent years: 2016 (black), 2017 (red),
2018 (green), and 2019 (blue) from January to May.

We examined the relationship between the first mode of the EOF and atmospheric forcing using a

time-lagged correlation. No clear correlation was discernible from this analysis (not shown). But in
comparison with the previously published observations from 2016 (Asfahani et al., 2020*)*, the period of
negative average heat flux lasted longer into the spring. Thus, there is an overall difference in the duration
of the negative heat flux between the 2016 observations and the observations in 2019.

The surface layers responded to the heat loss with decreased temperatures and increased salinity and

density. The cumulative effect of the cooling through the entire winter period resulted in the formation of
the densest surface waters in late March, when the difference in temperature and salinity between the
surface and the deep layers was at a minimum. Following this cooling phase, the net heat flux fluctuated
around zero. Then in early April, the weakening of the atmospheric forcing, the transition to positive heat
fluxes, and restratification due to advection from the south resulted in a near-surface temperature increase
of 1˚C and a salinity decrease of 0.2, both of which contributed to the near-surface density decrease in
April (**Fig. 4**). The warmer, less saline, and thus lighter water from the south spread into the area, and
during the restratification, it overrode the denser waters, isolating them from additional direct ocean-
atmospheric interaction.

The water underlying the more buoyant surface water, which lies along the 28.2 kg/m$^3$ isopycnal,

extended from the surface in mid-transect to approximately 200m nearest the coast (**Fig. 8**). This recently
exposed subsurface water spreads in the basin, and its signal can be detected in the CRS, as mentioned
by Zarokanellos and Jones (2021). This water results from the northward advection of GASW, which is
subjected to evaporation along its entire transit of the RS. Winter cooling throughout the entire period
further modifies the transported surface water in the NRS. During this particular year, the presence of the
AE in the NRS temporarily blocked advection from the south, contributing to the surface waters' extended
exposure to the atmosphere. Near the end of the cooling period, when the surface water reached its
maximum density, the cyclonic circulation was reestablished, contributing to the inflow of buoyant
GASW which overlaid the denser water.

In this study, the PWP model was applied to subsets of the observational period to further understand

the relationship between the local heat flux and the advection of water from the south. The PWP model
used daily surface heat flux and wind stress, as a pronounced diurnal cycle was not evident in the observed
salinity and temperature data. While this simple 1-dimensional model cannot capture the spatial
variability of the water column structure or the atmospheric forcing field, it effectively illustrates the role
of atmospheric forcing in driving the seasonal evolution of the mixed layer in the absence of these
complexities. In contrast, Krokos et al. (2022) used a 4-dimensional MIT-GCM model to investigate the
spatial and seasonal evolution of mixed layer variability across the entire RS, highlighting the critical role
of atmospheric forcing, particularly through its influence on mixed layer temperature. This broader
modeling approach supports the atmospheric-driven dynamics demonstrated by our PWP findings.

**Fig. 10** shows the evolution of the MLD in the upper 200 m based on the PWP simulation. Three

separate PWP simulations were performed, initiating each simulation at the onset of one of the three
phases determined from the in-situ observations. The initial temperature and salinity profiles for the model
run were taken from the glider section nearest the initiation point of the run. The cooling phase extended
from February 1st until March 8th (35 days). During this period, first presence of a mesoscale cyclonic

eddy and later an anticyclonic eddy in the study area were observed. In the same period, the simulated MLD is constantly deepening, reaching a maximum depth of 162m at the end of the cooling phase.

During the dense phase, the observed MLD shows large fluctuations and mismatches with the simulated MLD. The 1-D model failed to capture the shallowing of MLD during the dense phase. The observed discrepancy is based on the pycnocline depth that shallowed substantially, such that the dense pycnocline intersected with the surface. Lastly, the PWP-simulated MLD during the warming phase also shows a discrepancy with the observed MLD. During this phase, the observed MLD rapidly shallowed to less than 45 m, and the mean MLD was about 20 m. The shallowing of the MLD during the warming and freshening phases was attributed, in part, to the northward flux of the buoyant Gulf of Aden Water. But the shallowing of the pycnocline due to the cyclonic eddy also contributed to the shallowing of the MLD, as seen clearly in **Fig. 8**. In addition, the shallowing of the MLD due to the presence of the CE facilitates exporting the surface water masses to the deeper layers below the euphotic zone, as indicated from the backscatter (**Fig. 5 h**).

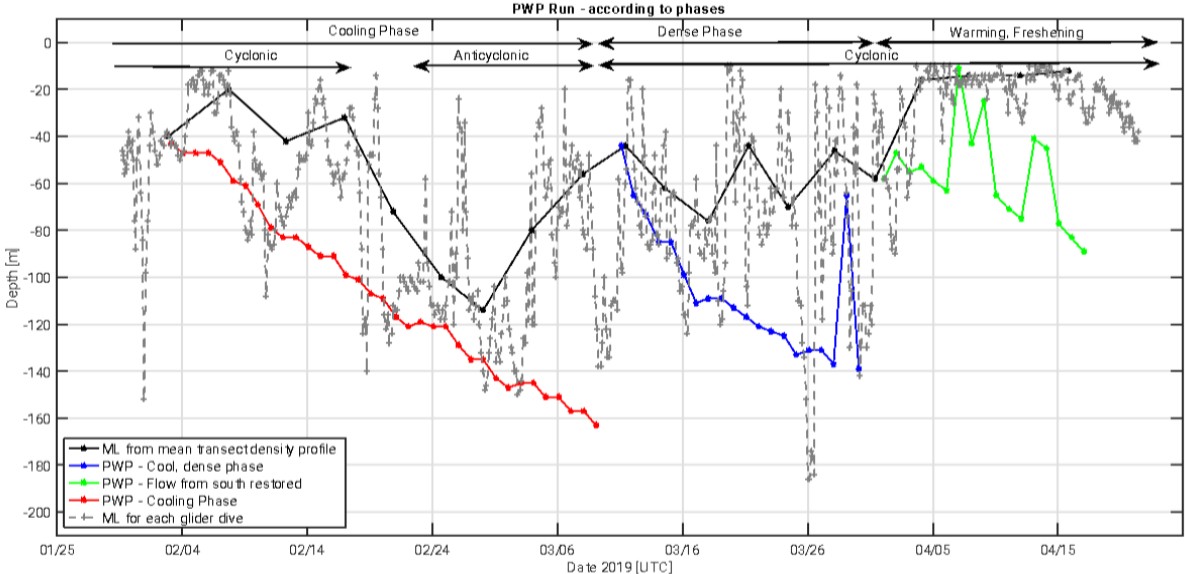

**Fig. 10:** Comparison between the observed transect-averaged MLD from the glider (black) and PWP-simulated MLD during the 1) cooling phase (red), 2) cool-salty and dense phase (blue), and 3) warming-freshening phase (green). The black + symbol shows the MLD for each dive, interconnected by the black dashed line.


Water mass subduction along isopycnals is a component of the formation of RSOW during winter
and a contributor to the carbon flux from the Zeu to the interior of the RS. As shown in **Fig. 5**, a water
mass containing elevated CHL and DO can extend well below the mixed layer and Zeu. Although only
one example with this feature (February 5-9, **Fig. 5**) is shown in this paper. Subducted water was evident
in the glider deployment from its deployment on January 30 through the fourth transect that was
completed on February 18. Key characteristics of this feature were elevated CHL and DO on the 28.2
kg/m$^3$ isopycnal that extended to as deep as 250 meters. In **Fig. 5**, this feature was evident from about 45
km offshore and shoreward. **Fig. 11** shows the relationship between these variables and density between
20 and 40 km offshore. A clear peak in both variables aligns with the 28.2 kg/m$^3$ isopycnal. Inshore, at
45 km, the region between 150 and 500 m contains a measurable fraction of the total CHL. In addition,
**Fig. 11d** shows that the subduction event contributes 20% to the integrated CHL between 150 and 500m.

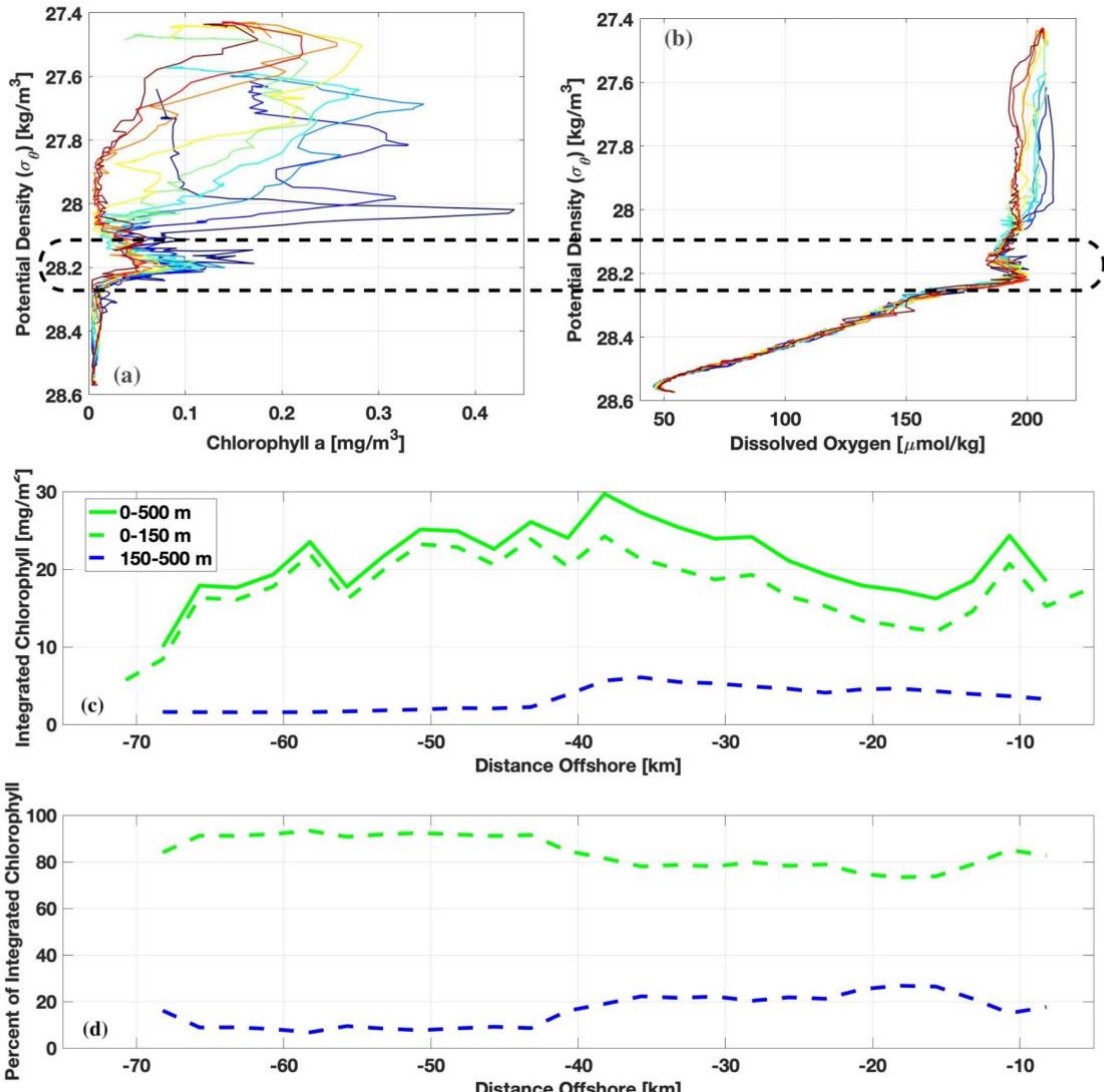


**Fig. 11:** Characteristics of the subducted feature in transect 2, February 4–9 (**Fig. 5**). Panels a and b show the concentrations of chlorophyll and dissolved oxygen as a function of density for profiles between 20 and 40 km offshore. Panel c shows the integrated chlorophyll as a function of distance offshore. The solid green line is the total integrated chlorophyll between the surface and 500 meters. The dashed green line is the integrated chlorophyll between the surface and 150 m, and the blue line is the integrated chlorophyll between 150 and 500 m. Panel d shows the percentage of the integrated chlorophyll from 0 to 150m (dashed green line) and from 150 to 500m (dashed blue line).

Given the limitations of our observations, constrained by the Exclusive Economic Zone (EEZ)
boundary, the full mechanism of the formation of this subducted layer is unclear. One possible mechanism
is that either vertical mixing or sinking of particles on the western half of the NRS (Kheireddine et al.,
2020) could create this feature, which is then entrained into the cyclonic circulation in this region and
transported from the western side of the basin to the eastern side. **Fig. 12** shows a conceptual diagram of
RSOW subduction and its biogeochemical impact on the NRS based on the existing observations.
Regardless of the details of the mechanism, subduction is a process that needs to be considered in the
physical and biogeochemical dynamics of the NRS.

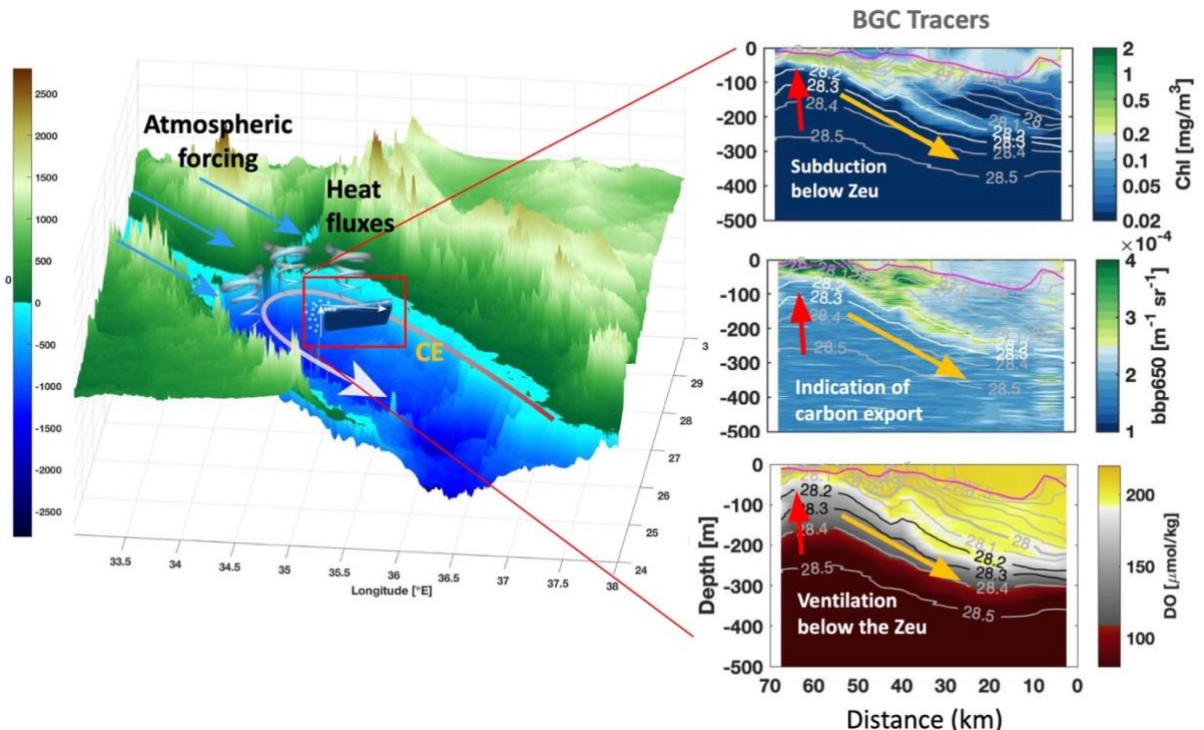

**Fig. 12:** Conceptual diagram of RSOW subduction and its biogeochemical impact on the NRS.
Topographic and bathymetric representation of the study area where physical and atmospheric processes
superimposed on the map indicate the general cyclonic circulation (grey arrow), the influence of
atmospheric forcing (blue arrows), and heat fluxes (grey spirals; left panel). Panels on the right display
the glider section of CHL (top), backscatter at 650 nm (middle), and DO (bottom). Red rows correspond
to the uplifting of the isopycnal and yellow arrows to the subduction of the newly formed RSOW.
Subducted CHL, backscatter at 650 nm, and DO are traced below the Zeu (~120 m) associated with carbon
export and ventilation of the deeper layers. The green isopleth of 180 μmol/kg presents the nitracline
(bottom-right panel).

## 5. Conclusion

The primary objective of this study was to understand the mechanisms contributing to the water mass formation of RSOW in the NRS and the associated biogeochemical responses. Our findings demonstrate that the RSOW formation is closely linked to the presence of a cyclonic eddy and intense winter cooling, consistent with previous studies (Asfahani et al., 2020; Sofianos and Johns, 2003; Papadopoulos et al., 2015; Yao et al., 2014b). A novel finding of this study is the role of water mass subduction, which, although not previously discussed, contributes not only to the RSOW formation but also to the carbon export as CHL and backscatter indicated that water has been subducted below the Zeu (**Fig. 12**). The observations also indicate that subduction events can significantly contribute to the ventilation of intermediate and deeper waters, thereby affecting the overall oxygen budget of the RS. During these events, oxygen-rich surface waters are transported into subsurface layers along the 28.2 kg/m³ isopycnal, facilitating oxygen redistribution at depth. The study also identified a transition from negative to positive heat flux and the re-establishment of northward flow along the eastern RS coast, signalling the cessation of the RSOW formation as less dense water from the south caps the denser northern waters. The presence of the AE south of the study area prevented the advection of more buoyant surface water. Although we could not determine the mechanism for this reversal of CE to AE from our observations and 1D model simulations. This likely influenced the observed water mass originating from the northern gulfs within the RSOW density range, consistent with Papadopoulos et al. (2015). This study highlights that multiple factors can contribute to RSOW formation (**Table 2**), including dense isopycnal surfacing from cyclonic circulation, vertical mixing, dense outflow from the northern gulfs, water mass subduction, the extension of dense isopycnal exposure due to blockage of buoyant flux from the south, and the eventual termination of these processes as the buoyancy flux is restored. In addition, it is clear that the submesoscale features are present in the region and contribute to the overall physical and biogeochemical dynamics of the region. To comprehensively capture the spatial and temporal dynamics of RSOW formation, future research should prioritize detailed observational and modeling studies, integrating autonomous platforms and ship-based sampling across the entire NRS basin. Such an approach would resolve the three-dimensional variability and provide valuable insights into the sources and sinks involved in RSOW formation and its biogeochemical impacts.

| Contributing Mechanisms | Yao et al 2014 (a & b) | Papadopoulos et al 2015 | Asfahani et al 2021 | Krokos et al 2022 | This Study |
|---|---|---|---|---|---|
| Cyclonic Circulation leading to exposure of dense isopycnals | √ | √ | √ | | √ |
| Convective mixing | √ | √ | √ | √ | √ |
| Upwelling/Downwelling along boundaries | √ | | | | |
| Outflow from Gulfs | | √ | | | √ |
| Along basin pressure gradients | √ | | | | |
| Submesoscale processes | | | | | √ |
| Subduction of dense water from surface water into pycnocline | | | | | √ |
| Anticyclonic blockage of northward flow of buoyant water into NRS | | | | | √ |
| **Biogeochemical effects** | | | | | |
| Eddy driven upwelling nutrient flux | | √ | √ | | √ |
| Convective mixing nutrient flux into euphotic zone | | | √ | | √ |
| Subduction results in the downward transport of dissolved oxygen and | | | | | √ |

| | | | | | |
|---|---|---|---|---|---|
| particulate carbon below the euphotic zone | | | | | |

**Table 2:** Summary of the major conclusions from the related studies relative to the formation of the

RSOW in NRS.

**Data availability**

The data sets that are presented in this paper (glider time series and gridded sections, 8-day remotely sensed SST, chlorophyll, and sea level anomaly/geostrophic velocity, and the NASA MERRA-2 reanalysis data) are available through the Zenodo repository  (https://doi.org/10.5281/zenodo.11046900)
https://zenodo.org/records/11046900?preview=1&token=eyJhbGciOiJIUzUxMiJ9.eyJpZCI6IjA4Z
DdkMzY0LTQyMGUtNGRiYS1iYTVmLWRkYjhlM2M2M2I2YyIsImRhdGEiOnt9LCJyYW5kb20iO
iI2MWM1YjUyOGQxZDBkYmRkOTc5MjI4MDYxZWEwMGJlZCJ9.VHVgGkzDwCZ1576vzMKco
mKl4Zax-uy9lbG_XHE1zfT_ag31O5DHNh0VHlJDWQNxfrn0S3HBbOXS_2QhoPTbDQ.

**Author contribution:**

All authors have contributed to Conceptualization, Data Curation, Formal Analysis, Investigation, Writing – original draft preparation and Writing – review & editing. Visualization is done by L.E., Z.K. and B.H.J.. Software is done by L.E. and B.H.J.. while supervision is done by Z.K. and B.H.J.

**Competing interests**

The authors declare that they have no conflict of interest.

**Acknowledgements**

The authors gratefully acknowledge the NASA Goddard Space Flight Center, Ocean Ecology Laboratory, Ocean Biology Processing Group for remote sensing data and the Copernicus website for the SLA data used in this study. The authors are grateful to the KAUST Coastal Marine Resources Core Lab (CMRCL) for their engineering and field support during the glider operations. Particular thanks go to Thomas Hoover, Samer Mahmoud, Mohammed A. Aljahdli and Lloyd Smith for their help with the glider deployments. The authors are also grateful to Dr. Luc Rainville for his suggestions and discussions regarding the PWP implementation. This research was supported by King Abdullah University of Science and Technology (KAUST) University. The ocean color products were obtained from NASA Ocean Color Group. The data are freely available online through the official website https://oceandata.sci.gsfc.nasa.gov/directdataaccess/Level-3%20Mapped/Aqua-MODIS/2019/.

The altimeter products were produced by Ssalto/Duacs and distributed by Aviso+, with support from Cnes (https://www.aviso.altimetry.fr). Dataset accessed [2022-01-27] at 10.5067/MODAM-8D4N9. The SST data source is NASA OBPG. 2020. MODIS Aqua Global Level 3 Mapped SST. Ver. 2019.0. PO.DAAC, CA, USA. MODIS CHL level 3 data was obtained via https://oceancolor.gsfc.nasa.gov/cgi/l3. We would like to acknowledge that the author Z.K. was partially part of the ITINERIS project.

**Financial support**

We thank the EU—Next Generation EU Mission 4 "Education and Research"—Component 2: "From research to business"—Investment 3.1: "Fund for the realisation of an integrated system of research and innovation infrastructures"—Project IR0000032—ITINERIS—Italian Integrated Environmental Research Infrastructures System—CUP B53C22002150006 for partially funding the author Z.K.

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
