# Peer review of "Mechanisms of the Overturning Circulation in the Northern Red Sea, more than Convective Mixing"

_EGUsphere, 2024_

## Author Comment (AC1)

**Manuscript: Mechanisms of the Overturning Circulation in the Northern Red Sea, more than Convective Mixing**

We deeply thank the reviewer for the constructive comments, which have significantly improved our manuscript. Below, we address each comment point by point (our responses are highlighted in blue).

**Reviewer 1**

Eyouni et al. present a nice overview of the various, complex mechanisms driving water mass formation in the northern Red Sea from a glider study occurring between January 31 to April 18, 2019. The authors do a nice job detailing many of the physical mechanisms from observations and model data as well as providing historical context to inform these findings. However, if the authors want to tie these physical findings to biogeochemical impacts in the region, further analysis and discussion is needed. I would recommend major revision but foresee this paper developing into a nice, comprehensive study. Below are my major and minor comments.

We thank Reviewer #1 for the constructive comments and suggestions. We revised the manuscript to add clarity and make better connections between the physical and biogeochemical processes in this manuscript. Specifically, we have revised and linked the biochemical background of the study area in the introduction (lines 62–85) and data and methods (lines 155–158 and Table 1). The results section has also been updated to consider backscattering (bbp) observations at 650 nm (when available; lines 363–371, 381-384, 421-422, 577–579). Furthermore, **Figs. 5–8** have been updated to reflect these changes. Finally, the discussion section now includes a more comprehensive and synthetic analysis of the impact of physical processes on the biogeochemical variability in the northern Red Sea. This analysis is illustrated by a conceptual diagram (**Fig. 12**) and detailed in lines 618-626 and 632-638.

Major Comments:
On lines 432-434, the authors write what I interpret to be a key argument of their study: "Regardless of the details of the mechanism, subduction is a process that needs to be considered in the physical and biogeochemical dynamics of the northern Red Sea." If the authors want to stress the biogeochemical significance of (all of) these physical processes, there needs to be more discussion (and actual chlorophyll and oxygen concentrations!) of the biogeochemical dynamics. The authors do not provide a biogeochemical background for the region. What limits primary production here, nutrients? light? A high salinity gradient? Does this region exhibit substantial or minimal primary production? What about carbon export? Mixed layer depth is discussed at length (great!), but the euphotic zone should be discussed in greater detail and included on figures, particularly if they relate to subduction.

We thank the reviewer for the comment. In response, we have revised the introduction to provide a more comprehensive overview of the biogeochemical context of the northern Red Sea (lines 62-85). Specifically, we have incorporated information regarding the depth of the euphotic layer (Zeu), which, based on estimates derived from vertical CHL distribution data and a CHL-dependent light attenuation model by Morel & Maritorena (2001) as implemented by Zarokanellos and Jones (2021), is located within the upper ∼120 meters. As noted by

Churchill et al. (2014) and consistent with the earlier work of Naqvi et al. (1986a), nitrate concentrations exceeding 1 µM/L are typically observed where oxygen (DO) concentrations fall below 180–185 µmol kg−1. Following Zarokanellos and Jones (2021), the 180 µmol kg−1 DO isopleth can indicate the nitracline. We adopted this approach in the conceptual figure (**Fig. 12**, lines 618-626) and explicitly stated it in our Discussion and Conclusions (lines 611-613 & 632-638).

Additionally, bbp should be included in the analysis. I outline below a number of places where actual concentrations should be provided and further explanation of the author's proposed biogeochemical impacts or changes is needed. I completely agree with the authors that glider profiles show important biogeochemical changes resulting from physical processes, but more discussion is needed to make the argument coherent enough for a general audience.

Revised as suggested accordingly in the results (lines 363–371, 381-384, 421-422), discussion (lines 577–579), conclusion (lines 632-638), and **Figs. 5-8 & 12**.

The various controls on water mass formation presented here are fascinating and I think readers would really benefit from a schematic showcasing these physical processes (the eddy transport of water, subduction, etc.).

We thank the reviewer for the suggestion. A schematic figure has been added to the manuscript (**Fig. 12**)

Could dive-averaged current (DAC; Frajka-Williams et al., 2011) be calculated from the glider data? It would make a nice addition when discussing the source of waters.

We thank the reviewer for the suggestion. We have revised the manuscript (**Figs. 5-8** and lines 388-389, 423-424,441-443, 473), and we have added at the end the depth average current, as there is a strong stratification in the study area, and averaging over a defined depth range can better capture the dynamics of the upper layer. Furthermore, while dive-averaged currents represent an integrated effect over the dive duration, depth-averaged currents provide a snapshot that can better represent the conditions at a particular time that we are interested in this study, especially capturing a transient eddy (**Fig. 6a**). In addition, the depth-averaged measurements are often more directly comparable with the SLA observations we use in **Fig 3**. We have added the DAC as a subplot in every selected transect (**Figs. 5a-8a**).

Minor Comments:

Line 91: was backscatter (bbp) measured? Presumably it was measured on the Wetlabs and could be included in the paper?

We thank the reviewer for the comment. The glider has a backscatter sensor measuring at 532, 650, and 880 nm. However, for simplicity, this study focuses solely on the 650 nm measurements, as our goal is to illustrate how physical processes influence other biogeochemical tracers during the formation and subduction of the RSOW. The data and methods section has been revised accordingly (lines 155-158), and Table 1 has been updated. **Figs. 5–6** have been updated and discussed in the results section (lines 363-371, 381-384, 421-422), while the bbp at 650nm was also included in the discussion (lines 577-579), conclusion (lines 632-635 and conceptual **Fig. 12**.

Line 94: what is the average depth of bottom here?

We thank the reviewer for the comment. Based on Emodnet bathymetric data (**Fig. below**), the average bottom depth within the glider's operational area is approximately 700 m.

[Figure]

Data source : Emodnet

**Fig**. shows the bathymetric data within the glider section.

Line 111: can the authors please clarify whether they did divide glider chlorophyll by 2 in accordance with Roesler?

Indeed, CHL values were divided by a factor of two; thank you for your comment. This has been clarified in the revised manuscript (lines 160–161).

Line 111: was any sort of quenching correction applied to the chlorophyll data?

We thank the reviewer for the comment. We found no evidence of significant quenching after an examination of the CHL profiles. The manuscript has been revised for clarity on this point (lines 160–162)

Figure 3: errors in the way panels d and h were printed/copied

Revised as suggested (updated **Fig. 3**)

Figure 3: could you please add a North arrow to one panel for orientation?

Revised as suggested accordingly in **Fig. 3.**

Figure 3: in the caption can you please clarify where sea level anomaly and geostrophic velocity data are coming from? ESA correct?

Capture has been updated; thank you for your comment (lines 186-188 and 307-308).

Lines 245-246: I think part of this sentence is missing? ". . .while the depth of 500 m has been selected [to represent the near bottom] because. . ."

We thank the reviewer for their feedback. The text has been rephrased accordingly (line 317).

Figure 4: in the caption be consist with how you refer to panel labels ("A" vs. "a")

Revised as suggested.

Line 278-279: can you please include average values or a range of chlorophyll and oxygen to give the reader an idea of how "elevated" they were? The oxygen is not particularly easy to evaluate based on the figure.

The reviewer's suggestion is valid; we included average values to provide clarity of the elevated CHL and DO along the isopycnal of 28.2 between 20 and 40 km offshore, within the 160–260 m (lines 357-371).

Line 279-280: can you please be explicit about why you presume the high chl and dissolved oxygen waters originated closer to the surface and were subducted downward along the isopycnal. I agree with you, but to someone without a strong biogeochemical background, it may not be 100% obvious as to why these waters had to be subducted and could not have generated elevated chlorophyll and DO at depth.

We appreciate the reviewer's comment, and we have added the following text in lines 381-393: The glider observations of CHL, backscatter at 650, and DO allowed us to independently trace the subduction with three different bio-optical tracers. Indeed, the observed elevated CHL is strongly linked to phytoplankton growth, which primarily takes place only offshore within the Zeu, where dense water ($\geq$28.1 kg/m3) rose to a depth shallower than 50 m, bringing up nutrients from deeper layers. Also, the Zeu is located around 120 m in the Red Sea, and light at greater depths is too low to sustain photosynthesis (Zarokanellos and Jones, 2021). Furthermore, this transient eddy about 43 km offshore was not observed in either the previous or the following section, and it was embedded within the larger-scale flow (**Figs. 5a, 5b,** and **5e**). The observed high DO concentration on the surface can be a result of photosynthesis. The co-occurrence of high CHL and DO at depths below the Zeu suggests that this water was originally at the surface before it transferred and subducted deeper. The fact that the high CHL and DO waters align along the 28.2 isopycnal (**Figs. 5, 11a**, and **11b**) indicates that their subduction is associated with an eddy wherein the denser surface water is forced below, with the lighter water following the 28.2 isopycnal rather than being vertically mixed.

Line 281-282: perhaps this definition should come before the first use of the word or incorporated into the sentence: ". . . at 20 km offshore, suggestive of subduction (i.e., transfer of fluid. . .)"

We agree with the reviewer and have revised the manuscript accordingly, addressing the term subduction in the introduction (lines 69–71).

Line 304-305: can you expand on the bolus of highly oxygenated waters? Were the chlorophyll concentrations high during this period as well? What are the average TS of the Gulfs of Aqaba or Suez?

We thank the reviewer for the suggestion and have included further information regarding the bolus in the manuscript (lines 408-428). We have included the TS diagrams of the Gulf of Aqaba water masses in our replies below, but we will not incorporate them into the manuscript as they are out of scope. Typically, the oxygenated waters are located in the surface layers within the MLD. However, the observed bolus indicated that high oxygenated waters had trapped below the MLD, between the 28.2 and 28.3 kg/m³ isopycnals at 150 to 250 m depths and between 20 and 50 km offshore. The average DO concentration within the bolus is ~177 µmol/kg, while CHL is around 0.0046 mg/m³. The surrounding waters below the 28.3 isopycnal indicate that the DO and CHL values reach 62 µmol/kg and 0.0029 mg/m³, respectively. Above the 28.2 isopycnals, the DO and CHL have values of 203 µmol/kg and 0.079 mg/m³, correspondingly. Compared to the underlying layers, CHL within the bolus is slightly elevated (~3.6%), while DO is significantly higher by approximately 285%. The thickness of the layer between these two isopycnals varies, ranging from less than 40 m, and the thickness of the trapped bolus is approximately 100 m, indicating a distinct water mass, which is also associated with low BVF. The observed elevated BVF around the bolus suggests that this is a stable water mass isolated from the surrounding water column rather than a result of vertical mixing. This lens is slightly warmer (~22.3°C) and more saline (~40.43) than other waters within the same isopycnal range along the transect (**Figs. 6c, 6d, 6f**). While its signature was not reflected in CHL (**Fig. 6g**), the bolus is also detectable in bbp (**Fig. 6h**), with a concentration nearly 11% higher than the surrounding waters (**Fig. 6h**). This bolus is likely outflow water from the Gulf of Aqaba, which might be advected into the region by the southward flow and subsequently captured and recirculated by the observed AE (**Fig. 6a**). Only a few studies are available regarding the water mass characteristics of the Gulf of Aqaba (Manasrah, 2002; Manasrah et al., 2004), suggesting that the upper 300 m of the Gulf exhibit conditions similar to those found in the upper 100 m of the NRS during winter, with temperatures ranging from 20.4°C to 22.4°C and for the salinity between 40.3 and 40.7.

[Figure]

| θ/S diagram from 21st February to 7 March 1999 (Manasrah et al., 2004) plus the bolus in red. | θ/S diagram is from a recent mission in 2021 from the dataset of Yasser et al., 2023 (DOI 10.17882/96463); the period is in the mid-summer period where |

[Figure]

| | strong stratification takes place, and the physical processes that take place are different (data collected on 29/06/2021). |

Figures 5 and 6: are your chlorophyll concentrations mg L-1 or µg L-1? I'm assuming this is a typo and should be corrected to either mg m-3 or µg L-1

We thank the reviewer for the suggestion and have revised the units in our manuscript to $mg/m^{-3}$.

Line 321-322: can you comment as to why the uplift of low DO and low CHL waters is potentially biogeochemically important? 100 m is still quite deep. Are you hypothesizing the waters will be uplifted further, into the euphotic zone, thus enabling phytoplankton to engage in primary production? Or are you hypothesizing more biomass rich waters could contribute to the grazing or remineralization structure of those more near shore waters?

Revised as suggested, and we have updated the manuscript in lines 449-460 as follows: The uplift of isopycnals affects the biogeochemical processes by bringing low DO and CHL waters into the Zeu. This process modulates nutrient, carbon, and DO availability and ultimately affects primary production. Phytoplankton growth depends on the nutrients and light availability. The low-CHL waters typically indicate nutrient-depleted conditions at the surface, while the low-DO waters in deeper layers are generally enriched with remineralized nutrients such as nitrate, phosphate, and silicate (Garcia H.E. et al., 2018). In this case, the low-CHL and DO waters have reached ∼60 m, penetrating the Zeu, which extends to ∼120 m, as reported by Zarokanellos and Jones (2021). When these nutrient-available waters reach the Zeu, they can stimulate phytoplankton blooms, enhancing primary production (Falkowski et al., 1998). Further, the uplift of the 28.3 isopycnal (∼60 m) due to the presence of the cyclonic eddy (**Fig. 7**) affects nutrient availability (Zarokanellos & Jones, 2021; Kurten et al., 2019). This mesoscale eddy activity in the region often drives the shift in the phytoplankton community (Kurten et al., 2019).

Lines 350-353: this is currently a very short paragraph. Can you expand on this or potentially combine it with the following paragraph?

The paragraph in lines 490-498 has been updated for clarity as follows: The NRS is a dynamic and complex three-dimensional circulation with significant seasonal variability influenced by strong atmospheric forcing through wind stress and air-sea buoyancy fluxes. Direct observations and modeling experiments have both captured the formation of locally produced intermediate (RSOW) and deep water (RSDW) masses and their interactions with adjacent gulfs of Suez and Aqaba (**Table 2**; Asfahani et al., 2020; Sofianos and Johns, 2003; Papadopoulos et al., 2015). Two main thermohaline cells are associated with water mass formation and influenced by mesoscale dynamics, wintertime cooling, and deep convection. Numerical simulation studies suggest that the cyclonic gyre is the most probable site for RSOW formation (Yao et al., 2014a; Sofianos and Johns, 2003).

Lines 364-365: "this negative phase is consistent with the period when the circulation was anticyclonic". To me, this seems like a noteworthy finding. Perhaps emphasize it more?

We thank the reviewer for the suggestion, and we have explained the role of the negative phase in 2019 where the circulation was AE with the following text (lines 514-520): Recently, a study by Mohamed and Skliris (2025) showed that the annual climatology of the sea level is generally higher on the eastern boundary of the RS compared with the western boundary, where many areas are isolated patches of higher or lower values of SLA that indicate mesoscale activity. The maximum values correspond with regions where AEs are present. In our study, the negative phase of the EOF analysis (**Fig. 9b**) aligns with the presence of AE in the NRS, which appears to block or at least redirect the northward flow as has also been previously observed in the Central Red Sea (Zarokanellos et. al., 2017).

Lines 397-399: The statement beginning "Krokos et al." could be tied to the authors' PWP finding better.

We thank the reviewer for the comment; we have revised the text in lines 551-560 as follows: 'The PWP model used daily surface heat flux and wind stress, as a pronounced diurnal cycle was not evident in the observed salinity and temperature data. While this simple 1-dimensional model cannot capture the spatial variability of the water column structure or the atmospheric forcing field, it effectively illustrates the role of atmospheric forcing in driving the seasonal evolution of the mixed layer in the absence of these complexities. In contrast, Krokos et al. (2022) used a 4-dimensional MIT-GCM model to investigate the spatial and seasonal evolution of mixed layer variability across the entire Red Sea, highlighting the critical role of atmospheric forcing, particularly through its influence on mixed layer temperature. This broader modeling approach supports the atmospheric-driven dynamics demonstrated by our PWP findings.'.

Line 401: Typo: "The"

Corrected as suggested, line 563

Line 420: "Water mass subduction is a component of the formation of Red Sea Outflow Water during winter and a contributor to the vertical carbon flux from the euphotic layer to the interior of the Red Sea". I think this is the first time carbon is really discussed. While I have no doubt this is true, if the authors want to stress this as part of why better understanding the formation of RSOW is important, there needs to be carbon data (or bbp700 or bbp460 which again, I assume was measured via the wetlabs). Chlorophyll and oxygen ≠ carbon

We thank the reviewer for the suggestion and have updated the whole manuscript based on the backscatter observations in the following lines (155-158, 363-371, 381-384, 421-422, 577-579, 618-626, 632-635).

Line 428: what does "a measurable fraction of total chlorophyll" mean?

We thank the reviewer for the comment. The term "a measurable fraction of total chlorophyll" was used to emphasise that the measured values represented only a subset of the total CHL present. As the total chlorophyll and chlorophyll fluorescence are not exactly the same. The total chlorophyll typically refers to the concentration of chlorophyll pigments measured via

extraction methods (such HPLC; Shioi Y., Fukae R., Sasa T. (1983)) and can provide the actual pigment concentration. In contrast, chlorophyll fluorescence is a non-destructive, in situ measurement that detects the natural fluorescence emitted by chlorophyll when they are excited by light. While fluorescence is often used as a proxy for chlorophyll concentration, it can be influenced by factors such as phytoplankton physiological status, light availability, and environmental conditions, meaning that the two measurements may not always match perfectly, especially in the Red Sea the phytoplankton community is not well studied.

Figure 11: axes labels are hard to read and look squashed. A-d labels look abnormally large relative to other figures. I appreciate the inclusion of panel d but I don't think it is discussed anywhere in the paper? Overall figure (export) quality could be improved.

We thank the reviewer's suggestion and have significantly improved the quality of **Fig. 11**. we have also added lines 596-597 & 605-606.

Table 2: I would suggest introducing Table 2 earlier, before the conclusion

We thank the reviewer for the suggestion. **Table 2** provides a concise summary of our key findings, helping readers synthesize the presented results. It also places our findings in the context of previous studies. Therefore, we anticipated and introduced **Table 2** in the Discussion section instead of the Conclusions session.

---

## Author Comment (AC2)

**Reviewer 2**

The proposed work deals with the formation of intermediate waters in the northern Red Sea (RSOW) during winter (2019), analyzing physical and biogeochemical mechanisms through glider observations, satellite data, and models. The authors explore the role of cyclonic vortices and atmospheric forcing in ventilation and nutrient transport. The work highlights key processes such as cooling, convective mixing, and mesoscale interaction in ocean dynamics. The paper presents interesting new features and, in general, is well written with a high quality of figures that allows for a quick and thorough understanding of the work. Despite this, some doubts remain about the description and motivation for the use of some datasets, and some minor considerations about the writing.

We thank the reviewer for their comments. We have revised the manuscript to provide clarity and explain in detail the datasets that have been used.

1) Recheck the introduction of acronyms, they are often not introduced properly, even if you add the citation, the acronym must be described explicitly. E.g. RSOW, AE etc, possibly with the acronym next to the extended form.

We thank the reviewer for the suggestion, and we have reviewed the whole manuscript for consistency.

2) The introduction is detailed with respect to some physical aspects of the studied area, but the topic of the work could be described more accurately, the state of the art of this topic with a quick overview of the motivations.

We thank the reviewer for the suggestion. We have revised and updated the introduction accordingly, focusing on the description of the mechanism of RSOW formation, including the coupling of the physical and biogeochemical processes in the study area.

3) Please emphasise the motivations of the study also in the conclusions.

We have revised the conclusion section, emphasising our motivations (lines 628-654)

4) Suggest putting the results obtained into more context with previous studies on this area.

We agree with the reviewer comment to contextualize our results. We have revised both the discussion and conclusion sections, providing a synthetic overview of the previous studies in the area. **Table 2** summarizes the major conclusions from the previous studies in the study region in relation to our study regarding the formation of the RSOW in NRS.

5) Why did you choose to use the MERRA dataset, even though there are better performing datasets with a higher resolution (e.g. ERA5)? I suggest justifying your choices and showing that MERRA-2 is the best choice, particularly for the basin in question. It would be interesting to show a very brief comparison MERRA-2 vs ERA5 vs FNL-GDAS (and/or analysis data from global models, such as IFS or GFS).

We thank the reviewer for the comment. We revised the manuscript in the lines 202-208. Our decision to use MERRA/MERRA-2 dataset was based on two factors. The first is based on

having consistency with previous studies in the Red Sea, so our work can be comparable. The second factor is that a dedicated study regarding the heat fluxes over the study area demonstrated that both MERRA-2 and ERA5 provide comparable and accurate heat fluxes in the northern Red Sea. We have cited the relative study of Al Senafi et al. (2019) [Surface Heat Fluxes over the Northern Arabian Gulf and the Northern Red Sea: Evaluation of ECMWF-ERA5 and NASA-MERRA2 Reanalysis - https://doi.org/10.3390/atmos10090504] in this manuscript. Based on their study, both ERA5 and MERRA2 provide accurate heat flux data in the northern Red Sea with a correlation of 0.97–0.98. Furthermore, both represent the seasonal variability and wind effects on air-sea fluxes accurately. A detailed comparison of different atmospheric datasets is beyond the scope of the present study.

6) Regarding the figures, I suggest enlarging the fonts and optimising the spaces between the panels

We thank the reviewer for the suggestion. The figures have been updated to improve clarity and increase the font size.

---

## Author Response (AR2)

Replies to reviewer 1:

We sincerely thank the reviewer for the helpful comments. We have addressed each point individually—please find our detailed replies below in blue.

Line 372-373: "observed elevated CHL is strongly linked to phytoplankton growth. . ." The authors did not actually measure growth, right? What we are talking about here is assumed growth? Please clarify.

'Indeed, the observed elevated CHL—commonly associated with phytoplankton growth in the literature—primarily occurs only offshore, within the Zeu…' (Line 372-373).

Throughout: Use scientific notation when necessary (CHL conc)

We have updated were was needed the CHL concentrations with scientific notation:
- CHL ($\sim$30 × 10⁻³ mg/m³)…(Line 348)
- CHL was $\sim$5 × 10⁻³ mg/m³ (Line 350)
- while CHL is around 4.6 × 10⁻³ mg/m³. (Line 401)
- waters below the 28.3 isopycnal indicate that the DO and CHL values reach 62 μmol/kg and 2.9 × 10⁻³ mg/m³, respectively. Above the 28.2 isopycnals, the DO and CHL have values of 203 μmol/kg and 79 × 10⁻³ mg/m³… (Line 402-404)

Lines 445 - 448: I am confused about the relationship between these sentences. Aren't they talking about the same mechanism- the uplift of nutrient rich waters along the 28.3 isopycnal? Perhaps it is just the use of "further" that confuses me?

Dear reviewer we have rephrased as you suggested. We agree that the use of 'further' may have caused some confusion. The revised sentence now reads: When these nutrient-available waters reach the Zeu, they can stimulate phytoplankton blooms, enhancing primary production (Falkowski et al., 1998). The uplift of the 28.3 isopycnal ($\sim$60 m) due to the presence of the cyclonic eddy (**Fig. 7**), also influences nutrient availability…(Lines 445-448)

Lines 485 - 487: Should this read: "Unlike previous observations and interpretations of the NRS, [this study observed] a reversal of the currents in the eastern half of the basin prevented the inflow of warmer, fresher water from the south" ?

You are absolutely right, and we have rephrased this accordingly.
Unlike previous observations and interpretations of the NRS (e.g., Asfahani et al., 2020; Papadopoulos et al., 2015; Yao and Hoteit, 2018), this study observed a reversal of the currents in the eastern half of the basin prevented the inflow of warmer, fresher water from the south.(Lines 485-487)

Furthermore, we have updated the colorbars in several figures (Figures 5–8 and Figure 11) to comply with colorblind-friendly design. Specifically, we have adopted **cmocean** colormaps to ensure perceptual uniformity and accessibility. (Thyng, Kristen, et al. "True Colors of Oceanography: Guidelines for Effective and Accurate Colormap Selection." Oceanography, vol. 29, no. 3, The Oceanography Society, Sept. 2016, pp. 9–13, doi:10.5670/oceanog.2016.66.)

---

## Author Response (AR3)

We thank the Editor for carefully reviewing our submission and highlighting these pending details.
Please find our point-by-point responses provided below in blue.
Additionally, all new corrections are highlighted in green in the revised manuscript.
We appreciate the effort invested in this review, which has undoubtedly improved the quality of our work.

The authors have addressed all the previous minor comments; however, a few additional small edits are required before the manuscript can be accepted for publication:

The abbreviation CHL is used throughout the manuscript, but I could not find where it is first defined. Please ensure that it is defined upon its first appearance in the text.

Dear Editor we have define now the CHL in line 19 of the updated manuscript.

- CTD and DO are defined in Table 1, but they are used earlier in the text. Please define them at its first mention.

Dear Editor we have define now the CTD in line 129 and the DO in line 130 of the updated manuscript.

- In the caption of Figure 12, you use the abbreviation CHL, but it is not used in Figure 11. Please modify the captions for consistency.

Dear Editor, we have updated the caption of Figure 11 in lines 582-588.

- The updated colorbars refer to Figure 12 and not Figure 11—is that correct? Please confirm and clarify if needed.

The updated colorbars correspond to Figure 12, not Figure 11, as was incorrectly stated.

---

## Author Response (AR4)

We thank the Editor for the decision and comments, which we have addressed accordingly. Our replies are provided in blue.

I recommend that the manuscript be considered for publication after the following comments have been addressed. I do not require a review of the revised version.

- As CHL is mentioned only twice in the abstract, I suggest avoiding the use of the abbreviation. Instead, please define it in line 19 of the Introduction section.

We have removed the abbreviation from the Abstract and inserted it at line 67, the place where for the first time chlorophyll is mentioned in the Introduction section.

- Kindly revise line 130 to read: dissolved oxygen (DO) sensor.

We have revised it accordingly.